# Arctic zircon U-Pb ages reveal multiphase glaciations in East Siberia during the late Quaternary

Han Feng [1,9], Zhengquan Yao [1,2,3,9], Xuefa Shi [1,2,3 ✉], Zhongshi Zhang [4 ✉], Huayu Lu [5], Hanzhi Zhang[5], Yanguang Liu [1,2,6], Xin Shan [1,2], Jiang Dong [1,2], Linsen Dong[1,2], Gongxu Yang[1], Limin Hu[2,7], Yuri Vasilenko [8], Anatolii Astakhov [8] & Alexander Bosin [8]

Tracing ice-rafted debris (IRD) in Arctic Ocean sediments is crucial for understanding the evolution of Northern Hemisphere ice cover. However, uncertainties in identifying the provenance of IRD across circum-Arctic shelves have complicated reconstructions of the East Siberian Ice Sheet (ESIS). Here, we present a provenance study using 10,111 detrital zircon U-Pb ages from circum-Arctic shelf sediments and central Arctic IRD. Our results reveal that a prominent ~90–110 Ma zircon peak serves as a distinctive fingerprint for East Siberian-sourced sediments. Central Arctic IRD from at least four glacial periods contains this signature, indicating repeated ESIS glaciation, likely within the past three glacial cycles. This multiphase glaciation of East Siberia exerted significant, though poorly understood, influences on both polar and global climates during the late Quaternary. Our findings provide valuable insights into the history of Northern Hemisphere glaciation and offer an effective approach for reconstructing ice sheet evolution.

The provenance of ice-rafted debris (IRD) in deep-sea sediments of the Arctic Ocean provides critical insights into the evolution of Northern Hemisphere ice cover[1–7]. Ice sheets across the Arctic produce large amounts of IRD that are subsequently transported by icebergs[8]. As icebergs melt, they release their IRD load, depositing it on the ocean floor[9]. Consequently, the iceberg-carried IRD potentially contains unique provenance signals that reflect erosion and sediment production by different ice sheets on the Arctic continents and continental shelves (Fig. 1).

The Arctic Ocean has received substantial IRD influxes during past glacial-interglacial cycles[8]. However, identifying the precise sources of these IRD remains challenging due to the difficulty of distinguishing debris provenance across the circum-Arctic continental shelves (Fig. 1). Provenance proxies, such as mineral assemblages and isotope geochemistry, have been effective in differentiating sediment sources between the continental shelves of North America and the entire Eurasia (Fig. 1 and Supplementary Fig. S1)[1,10,11]; however, these methods struggle to distinguish detritus from the continental shelves of eastern Eurasia (i.e., East Siberia) versus western-central Eurasia (Supplementary Figs. S1, S2 and S3; see details in Supplementary Text S1). Although Fe oxide indicators have proven capable of identifying East Siberian sources[12], their

[1]Key Laboratory of Marine Geology and Metallogeny, First Institute of Oceanography, Ministry of Natural Resources, Qingdao, China. [2]Laboratory for Marine Geology, Qingdao Marine Science and Technology Center, Qingdao, China. [3]Key Laboratory of Deep Sea Mineral Resources Development, Shandong (preparatory), Qingdao, China. [4]Department of Atmospheric and Oceanic Sciences, School of Physics, Peking University, Beijing, China. [5]Frontiers Science Center for Critical Earth Material Cycling, School of Geography and Ocean Science, Nanjing University, Nanjing, China. [6]College of Ocean Science and Engineering, Shandong University of Science and Technology, Qingdao, China. [7]College of Marine Geosciences, Key Laboratory of Submarine Geosciences and Prospecting Technology, Ocean University of China, Qingdao, China. [8]V.I.Il'ichev Pacific Oceanological Institute, Far Eastern Branch of Russian Academy of Sciences, Vladivostok, Russia. [9]These authors contributed equally: Han Feng, Zhengquan Yao. ✉e-mail: xfshi@fio.org.cn; zhongshi.zhang@pku.edu.cn

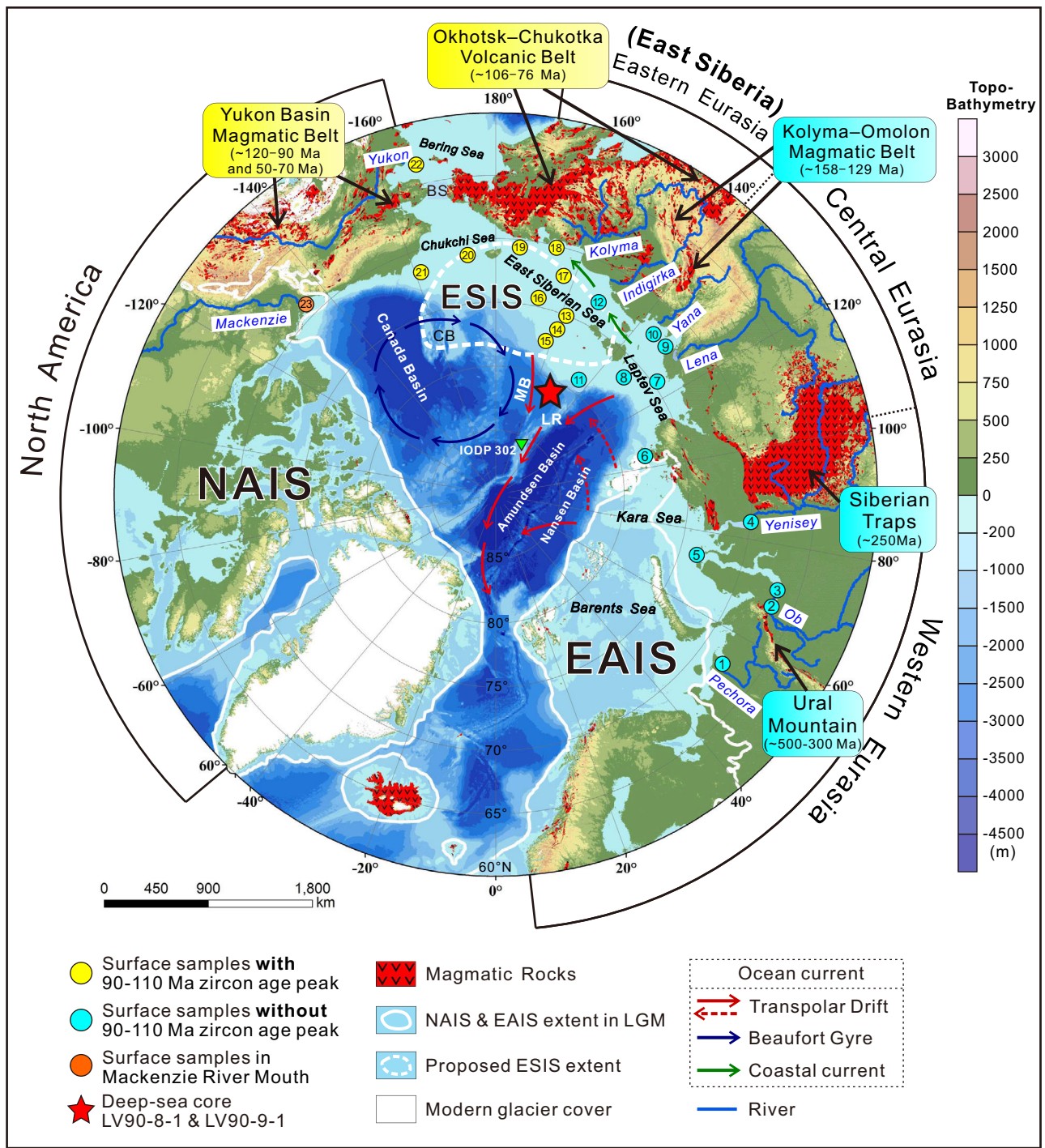

**Fig. 1 | Regional setting of the circum-Arctic region.** The outer circle presents range of the circum-Arctic continental shelves, where the entire Eurasian continental shelf is divided into three parts: western, central, and eastern Eurasian (East Siberia) continental shelf. NAIS (North American Ice Sheet) and EAIS (Eurasian Ice Sheet) during LGM (Last Glacial Maximum) are enclosed by the white solid line[79]; Proposed ESIS (East Siberian Ice Sheet) is enclosed by the white dashed line[19]. The times of intrusion or eruption of typical magmatic belts[80] are marked by black arrow with notes in colored rectangulars[27]. The colors of the surface sample points in this figure correspond to the fill colors of the zircon age distributions shown in Fig. 2. The red solid/dashed arrows indicate possible direction of Transpolar Drift[6]. Topographic and bathymetric data are from the GEBCO Grid (https://www.gebco.net/data-products/gridded-bathymetry-data). LR: Lomonosov Ridge; MB: Makarov Basin; CB: Chukchi Borderland; BS: Bering Strait.

application has not revealed a significant East Siberian signature in Arctic Ocean sediments[13].

This absence of a clear East Siberian signal in the sedimentary record has limited our understanding of Northern Hemisphere glaciation dynamics during past glacial-interglacial cycles. Previous provenance studies of IRD have identified iceberg surges originating from ice sheets on Eurasian or North American continental shelves in the central and eastern Arctic Ocean (including the Lomonosov Ridge, Makarov Basin, Amundsen Basin and Nansen Basin; Fig. 1) during the Quaternary glacial periods[2,4,5,7,14]. In the western Arctic Ocean (including the Canada Basin and the Chukchi Borderland; Fig. 1), sediments have received substantial quantities of IRD from the North American Ice Sheet via the

Beaufort Gyre[1,6,15,16]. However, the lack of evidence for an East Siberian contribution to IRD continues to hinder the reconstruction of glacial ice extent in this region. Whether and when an East Siberian ice sheet existed has remained a topic of debate for over two decades (Fig. 1)[17–20].

To address this question, we employed detrital zircon U-Pb age distributions as a provenance tracer to identify the source of Arctic IRD. We first analyzed surface sediments from circum-Arctic continental shelves to establish discriminative source signatures, and then we applied this provenance method to coarse-grained IRD in deep-sea cores from the southern Makarov Basin (central Arctic Ocean). Zircon grains are highly resistant to weathering, and their U-Pb age distributions in sediments are closely associated with the magmatic history of their source regions[21]. Although zircon grains are less abundant in carbonate and mafic rocks[22], they are prevalent in felsic and intermediate igneous rocks that are widely exposed across various Arctic source regions (Fig. 1). Compared to mineral assemblages and geochemical isotopes (e.g., Sr-Nd isotopes)[10,11], the age distribution of detrital zircons in sediments has the potential to accurately distinguish source regions with relatively small age differences (e.g., a few tens of millions of years).

## Results

### Exclusive provenance indicator for East Siberia

We present a dataset of detrital zircon U-Pb age distributions from surface sediments across the circum-Arctic continental shelves (Fig. 2). New U-Pb ages of 3705 detrital zircons in sand-sized particles (> 63 μm) were obtained from 17 surface samples collected from the marginal seas of the Eurasian continental shelf (Fig. 1), including the Chukchi Sea, East Siberian Sea, Laptev Sea, and Kara Sea (see "Methods"; Fig. 2 and Supplementary Table S1). Additionally, we compiled the published zircon ages of six samples from the Barents Sea (Pechora River mouth)[23], Kara Sea[23,24], North American continent (Mackenzie River mouth)[25] and Bering Sea (Yukon River mouth)[26] (Fig. 2; see sample locations in Fig. 1).

The zircon age distributions in these surface samples show regional distinctions (Fig. 2). Notably, a distinctive zircon age peak at ~90–110 Ma clearly fingerprints the East Siberian provenance (Fig. 2m-u) (provenance of other zircon age peaks are discussed in Supplementary Text S2). This age peak is found in sediments from the East Siberian continental shelf (including the East Siberian Sea and Chukchi Sea) but absent in sediments from the western-central Eurasian (including the Barents Sea, Kara Sea and Laptev Sea) and North American continental shelves, highlighting the difference between these areas (Fig. 2). The ~90–110 Ma zircons are likely derived from the Okhotsk-Chukotka Volcanic Belt (~106–76 Ma) that is exposed in East Siberia (Fig. 1)[27]. Although sediments from the Bering Sea also contain a 90–110 Ma peak (Fig. 2v), their unique ~60 Ma age peak−likely associated with magmatic belts in the Yukon River basin (Fig. 1)[27]−distinguishes them from those of the East Siberian continental shelf (Fig. 2m-u). Therefore, an age distribution characterized by a peak at ~90–110 Ma, with the absence of a ~60 Ma peak, serves as an exclusive provenance indicator for East Siberia, distinguishing it from western-central Eurasia, North America, and the Bering Sea.

### Sedimentary contribution from East Siberia to the deep Arctic Ocean

Two adjacent cores, LV90-8-1 (80.72°N, 152.50°E, water depth 2300 m) and LV90-9-1 (80.98°N, 152.46°E, water depth 2546 m), were collected from the southern Makarov Basin in the central Arctic Ocean (Figs. 1 and 3). Similar to most sediment cores from the Arctic Ocean, the brownish/reddish (higher a*, redness) layers in cores LV90-8-1 and LV90-9-1 are enriched in manganese (Mn) from circum-Arctic rivers, interpreted to represent an oxidizing environment with reduced sea ice during interglacial periods. In contrast, the grayish/yellowish (lower a*) layers indicate anoxic conditions, due to more extensive sea ice cover during glacial periods (Fig. 3a, d)[28].

The chronologies of LV90-8-1 and LV90-9-1 cores are based on AMS 14C dating of total organic carbon (TOC)[29] and lithostratigraphic correlations to adjacent well-dated cores (PS2757-8 and 29-GC1) on the Lomonosov Ridge (see "Methods"; Fig. 3). Earlier studies have dated PS2757-8 and 29-GC1 cores to MIS (marine isotope stage) 6[30,31], while recent studies using excesses in U-series isotopes ($^{230}Th_{xs}$, $^{231}Pa_{xs}$) have challenged this age model, extending it back to MIS 8[32,33]. Given the uncertainties in the age models, we put forward two alternative age models for LV90-8-1 and LV90-9-1 with marine isotope stages (Fig. 3; See details in Methods). Six glacial periods were identified in cores LV90-8-1 and LV90-9-1 (Fig. 3), following the correlation with cores PS2757-8 and 29-GC1[30,31]. The boundaries between glacial and interglacial periods were further refined based on the a* values (see "Methods").

In cores LV90-8-1 and LV90-9-1, sand contents remain low (< 5%) during interglacial periods (IGP) but increase significantly (>10%) towards the end or after each glacial period (GP) (Fig. 3c, f). An exception occurs during the late IGP 2 (i.e., late MIS 3), which exhibits a high sand content (~10%), possibly due to colder conditions during this period compared to MIS 5 and the Holocene in the Arctic Ocean[16]. The sand-rich layers deposited during late GP 3 and late GP 5 show a grain-supported texture with thin laminations and sharp basal surfaces (Supplementary Fig. S4), suggesting a turbiditic origin. In contrast, the sand-rich layers during GP 1, 2, 4 and 6, as well as late IGP 2, show a matrix-supported texture and gradational contacts with bioturbation (Supplementary Fig. S4); these attributes point to slower deposition rather than turbidite deposition. In addition, isolated granules and pebbles are found in these sand-rich layers (Supplementary Fig. S4), further suggesting that these sand particles are IRD. The MARs (mass accumulation rates) of IRD in these sand-rich layers range from 0.05 to 0.44 g/cm²/kyr (Supplementary Fig. S5), which is comparable to the range of IRD MARs (0.02–0.36) observed in the nearby IODP (Integrated Ocean Drilling Program) core 302[34] (Fig. 1).

Our detrital zircon U-Pb age analyses show that some sediment samples in sand-rich layers, four from core LV90-8-1 and two from core LV90-9-1, include the ~90-110 Ma age peak in age distributions (Fig. 4; see "Methods"). The ~90-110 Ma age peak occurs in six samples from four sand-rich IRD layers in late GP 1 (Fig. 4a), post GP 2 (Fig. 4c), late GP 4 (Fig. 4e, f) and late GP 6 (Fig. 4h, i). These six samples provide strong evidence of IRD contributions from the East Siberian continental shelf (Fig. 2m-u). However, the other two sand-rich IRD samples from late IGP 2 (Fig. 4b) and middle GP 6 (Fig. 4j), as well as two turbidite samples from late GP 3 (Fig. 4d) and late GP 5 (Fig. 4g), do not show ~90-110 Ma age peak. The age distributions of these four samples resemble those from the western-central Eurasian continental shelf surface samples (Figs. 2 and 4k), suggesting possible contributions from the Laptev Sea or the Kara Sea (Supplementary Text S2 and Fig. 2a–k). In addition, there are no age peaks at ~60 Ma in all core samples (Fig. 4a–j), suggesting a negligible contribution from the Bering Sea (Fig. 2v).

## Discussion

Arctic IRD can be transported by either icebergs or sea ice[8,35]. However, sea ice seems unlikely to be the main transporter of sand particles in the IRD layers of cores LV90-8-1 and LV90-9-1 (Fig. 3), especially for coarser particles (> 250 μm). The grains in these IRD layers are much coarser than those found in typical modern sea ice sediments[36,37]. Although sea ice can incorporate seabed particles > 63 μm through the anchor ice mechanism[38,39], previous studies have shown that modern sea-ice sediments seldom contain grains larger than 250 μm[37,39,40]. This is likely because sea ice primarily incorporates sediments from shelf areas away from the beach or nearshore zone, where the proportion of > 250 μm grains is typically less than 2% (Fig. 5). Even during glacial periods with lower sea levels, the coarse fraction on the shelves remained as low as it is today (Supplementary Fig. S6). In contrast, over

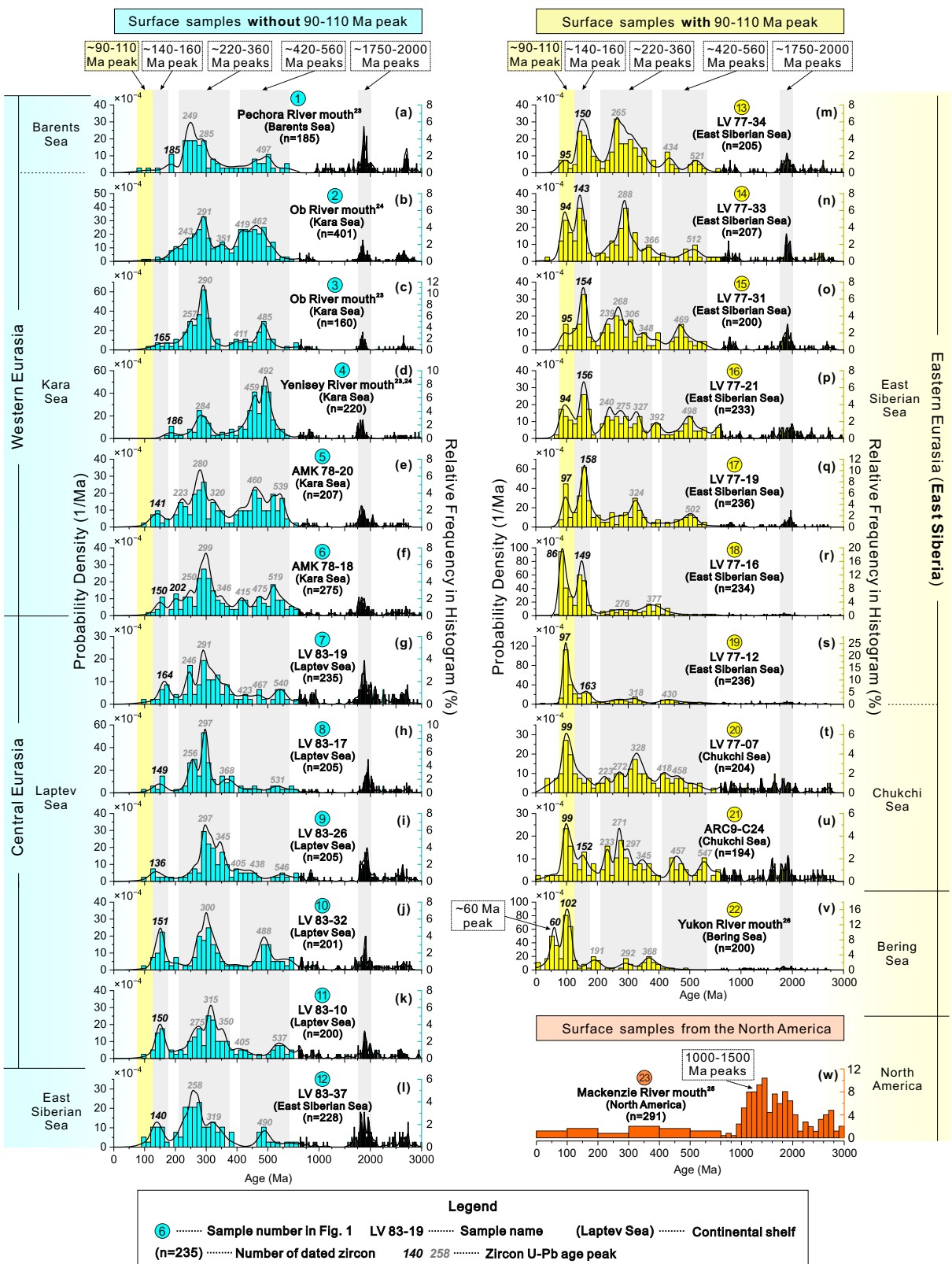

**Fig. 2 | Detrital zircon U-Pb age distributions of surface samples from circum-Arctic continental shelves.** Sample locations are shown in Fig. 1. Zircon U-Pb age distributions are illustrated by kernel density plots (KDE) and frequency histograms (see "Methods"). -90–110 Ma, -140-160 Ma, -220–360 Ma, -420–560 Ma, and -1750–2000 Ma zircon age peaks are shaded. Each part of the circum-Arctic continental shelf has a distinct zircon age distribution. The western and central Eurasian samples (**a–k**) lack the -90–110 Ma peak which is prominent in the East Siberian samples (**m–u**). The Bering Sea sample (**v**) has a unique -60 Ma peak. The North American sample (**w**) is characterized by 1000–1500 Ma zircons that are absent from all the Eurasian samples (**a–u**). Hence, the -90–110 Ma age peak exclusively fingerprints the East Siberian provenance. Source data are provided in Supplementary Data S2.

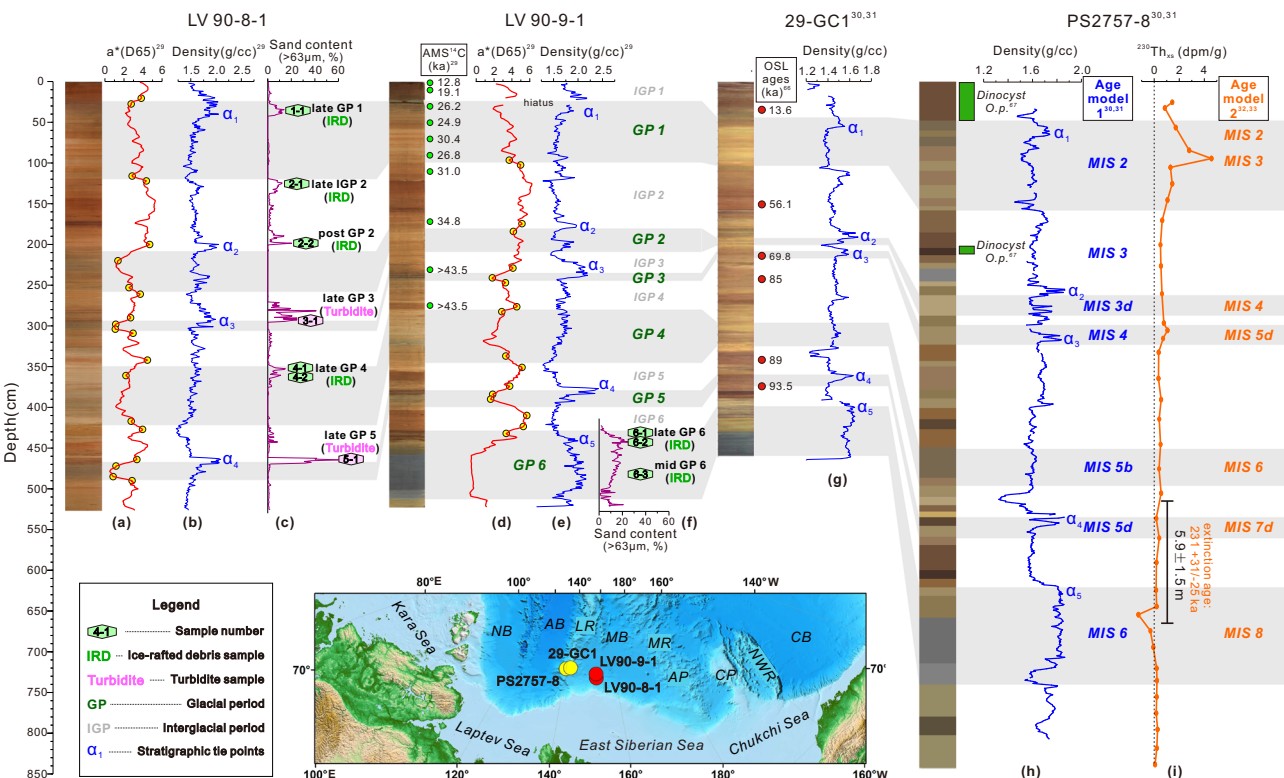

**Fig. 3 | Age model and sampling positions of detrital zircon for cores LV90-8-1 and LV90-9-1.** The color reflectance (a*; **a**, **d**)[29] of LV90-8-1 and LV90-9-1 were processed with a 5-cm moving average, where yellow points denote selected markers for glacial−interglacial boundary determination (see "Methods"). Glacial periods 1 to 6 are shaded and indicated in green italics. The chronology of core LV90-8-1 and LV90-9-1 is based on AMS 14C dating (green dots) on total organic carbon[29] and correlation to adjacent cores PS2757-8 and 29-GC1[30,31] from the Lomonosov Ridge. Bulk density stratigraphic tie points α1 to α5 from core PS2757-8 and 29-GC1[30,31] (**g**, **h**) are identified in core LV90-8-1 and LV90-9-1[29] (**b**, **e**), indicating good correlation between these cores. Red dots with age text in core 29-GC1 are optically stimulated luminescence (OSL) dating results[66]. The occurrence of dinocyst Operculodinium centrocarpum from PS2757-8 is marked by green bars, which allowed identification of MIS (marine isotope stage) 3 and

MIS 1[67]. The variations in 230Thxs for core PS2757-8 (**i**) are shown with its "230Thxs extinction age" (~231 ka, early MIS 7) at depth of ~5.9 ± 1.5 m[32,33]. The blue italic "MIS" refers to the age model from refs. 30, 31, while the orange italic "MIS" refers to the age model from refs. 32,33 (see "Methods"). Sediments with high sand content (usually 10%; **c**, **f**) were sampled for detrital zircon U-Pb dating, the postions of which are marked by hexagons with sample names. The thumbnail shows locations of these cores. Basemap of the thumbnailis from the ETOPO Global Relief Model (https://www.ncei.noaa.gov/products/etopo-global-relief-model). NB Nansen Basin, AB Amundsen Basin, LR Lomonosov Ridge, MR Mendeleev Ridge, CB Canada Basin, MB Makarov Basin, AP Alris Plateau, CP Chukchi Plateau, NWR Northwind Ridge. Source data of sand content are provided in Supplementary Data S3.

5% (~ 5–42%) of the sand in the IRD layers exceeds 250 μm (Fig. 5, and Supplementary Figs. S5 and S7), suggesting a different transport mechanism−most likely icebergs.

Moreover, the much coarser zircon grains found in the sand-rich IRD layers of cores LV90-8-1 and LV90-9-1, compared to surface samples from the surrounding continental shelves (Fig. 6), provide further evidence against sea ice as their transporter. Zircon, being one of the densest minerals, is more difficult to transport by rivers or currents than lighter minerals such as quartz or feldspar[41]. Considering that even lighter minerals >250 μm are rarely transported to the shelf (Fig. 5), theoretical estimates suggest that the equivalent threshold for zircons is ~150 μm (Supplementary Text S3), as shown by the absence of zircons >150 μm in modern surface sediments (Fig. 6). Therefore, sea ice, which incorporates seafloor particles through the anchor ice mechanism, lacks zircons larger than 150 μm. In contrast, ~6% to ~18% of zircons in the IRD layers exceed this size (Fig. 6 and Supplementary Fig. S7), implying that these coarse and dense grains likely originated from more energetic transport processes−most plausibly iceberg rafting.

In addition, scanning electron microscope (SEM) secondary electron images of quartz grains can be used to distinguish between iceberg and sea-ice origin sediments[42,43]. The surface microfeatures of quartz grains from a Holocene sea-ice origin sample (LV90-8-1 0-

3 cm) are subrounded with few mechanically-induced structures (Supplementary Figs. S8 and S9; further details in Supplementary Text S4). In contrast, quartz grains from sand-rich IRD samples are typically subangular and display numerous mechanical micro-features, including characteristic subglacial microfeatures including step-like fractures and subparallel linear fractures[44,45] (Supplementary Fig. S9). More than 60% of the quartz in the IRD layer show these subglacial microfeatures (Supplementary Fig. S9). While quartz grains in periglacial environments can also exhibit mechanical features[44], they lack these typical subglacial microfeatures observed here. These subglacial microfeatures indicate a signature of subglacial abrasion and crushing[42–45].

The sand particles in sand-rich IRD layers in cores LV90-8-1 and LV90-9-1 (Fig. 3) are also unlikely to have been supplied by fluvial processes, turbidity or bottom currents. Typically, rivers deposit and disperse sands in river mouths and along continental shelves, and further transport to the deep sea requires either turbidity currents[46] or ice-rafting[8,35]. However, the locations of cores LV90-8-1 and LV90-9-1 are more than 1000 km away from modern river mouths (including the Indigirka, Yana, and Lena Rivers; Fig. 1). Even if the sea level fell by ~120 m during the Last Glacial Maximum (LGM)[47], the site would still have been more than ~200 km away from river mouths, with a paleo-water depth of more than 2000 m. These preclude the direct influence

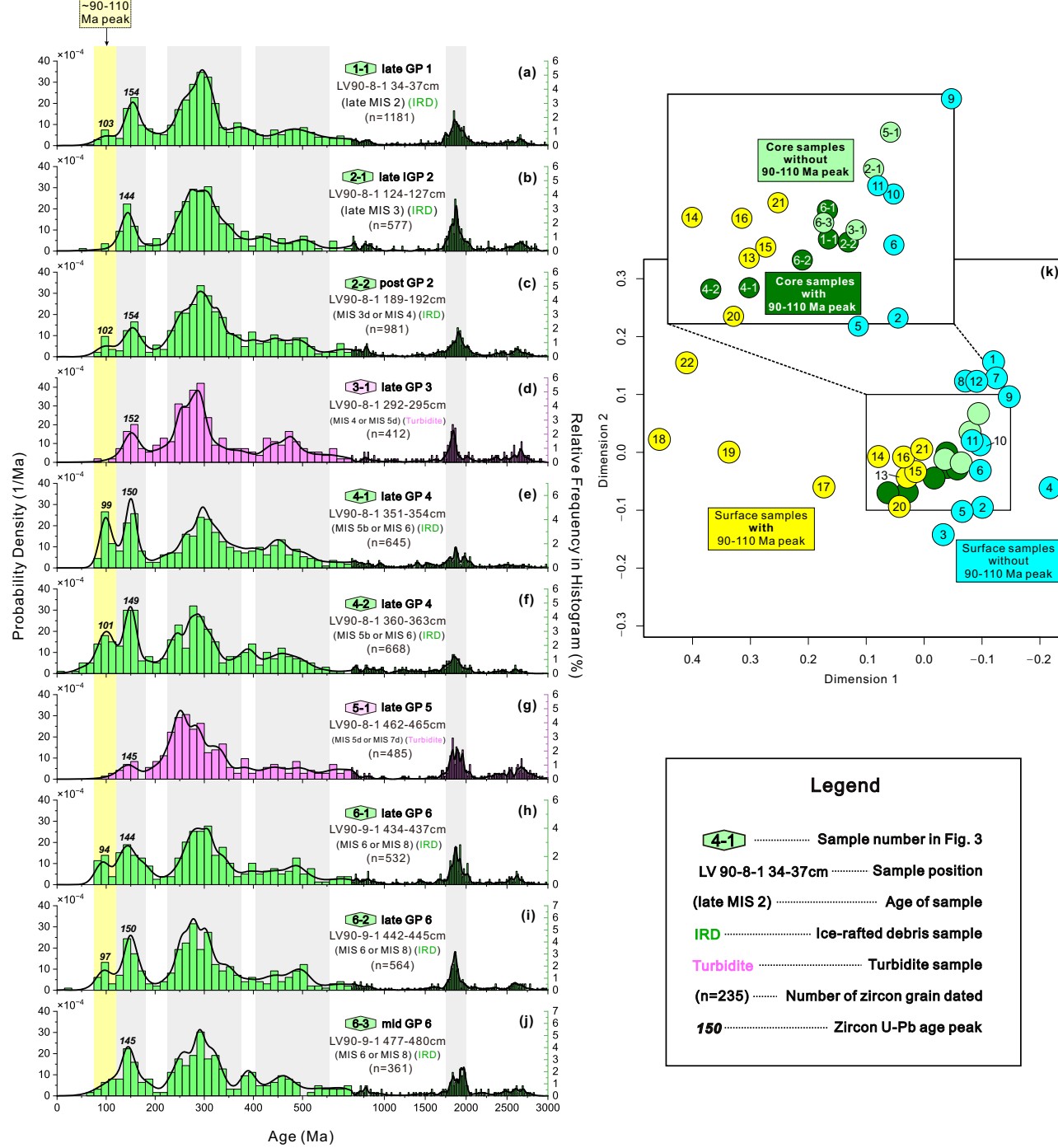

**Fig. 4 | Detrital zircon U–Pb age distributions of sand-rich layers from cores LV90-8-1 and LV90-9-1, with multidimensional scaling (MDS) plots comparing core and surface samples.** Sample positions are marked in Fig. 3. Zircon U-Pb age peaks of ~90-110 Ma, ~140-160 Ma, ~220-360 Ma, 420–560 Ma and ~1750-2000 Ma are shaded in the zircon age distribution. The age peak of ~90–110 Ma occurs in the ice-rafted debris (IRD) samples of late GP 1 (a), post GP 2 (c), late GP 4 (e, f) and late GP 6 (h, i), suggesting East Siberian origin. ~90–110 Ma zircons are scarce in the IRD samples from late IGP 2 (b) and middle GP 6 (j), as well as in the two turbidite samples from late GP 3 (d) and late GP 5 (g). Sample numbers of surface and core samples in the MDS plots correspond to those shown in Figs. 1–3. MDS plots were generated in (https://isoplotr.es.ucl.ac.uk/). Source data are provided in Supplementary Data S2.

of riverine processes on the sediments in these cores. More importantly, although turbidity events did occur during late GP 3 and late GP 5 (Fig. 3), the sand-rich IRD layers do not exhibit any of the features of turbidites (Supplementary Fig. S4). Additionally, to assess the role of bottom currents in transporting these particles, we examined the relationship between mean grain size and the percentage of sortable silt (10–63 μm)[48]. The observed relationship for samples in the sand-rich IRD layers, with an R value < 0.5 and a slope value < 0.2, does not

support the influence of bottom currents[48] (Supplementary Fig. S10). Moreover, the estimated bottom current velocity at the core site (< 5 cm/s) is insufficient for transporting coarse particles[49].

Therefore, we propose that iceberg was the most likely transport mechanism for the sand-rich IRD of LV90-8-1 and LV90-9-1, particularly for coarse sands > 250 μm and zircon grains >150 μm. The ~90–110 Ma age peak is prominent in the age distribution of coarse zircons >150 μm in sand-rich IRD layers (Supplementary Fig. S11), consistent

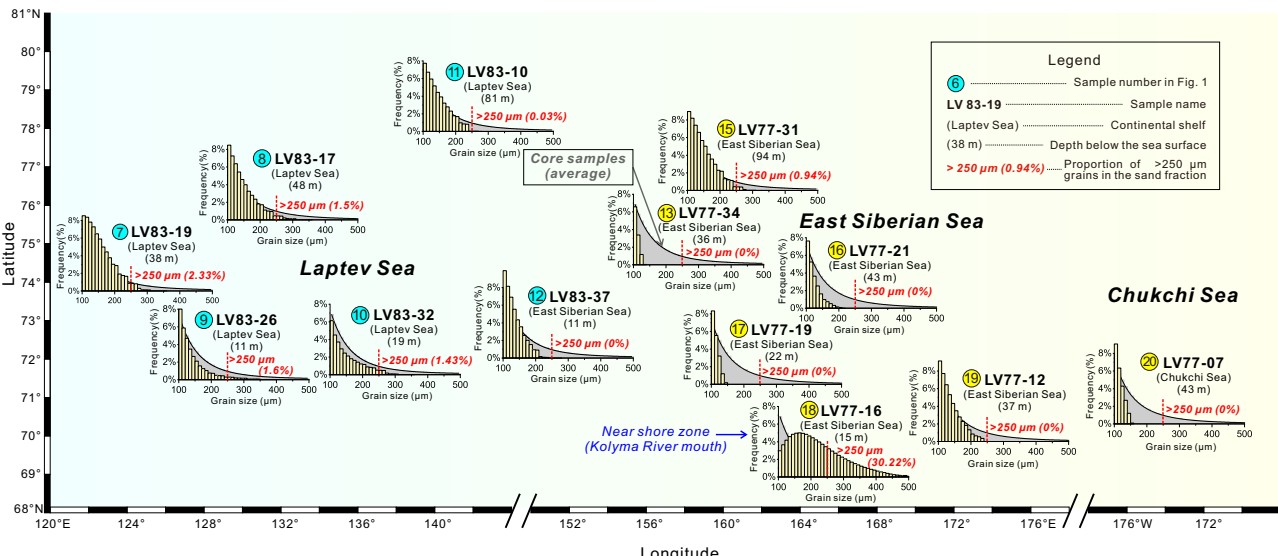

**Fig. 5 | Grain size distribution of 100-500 μm grains from the surface samples on the Laptev Sea, East Siberian Sea and Chukchi Sea.** The vertical scale shows frequency per 10 μm bin normalized to total sand fraction (> 63 μm). The gray histogram in the background of each sample represents the averaged grain size distribution from all sand-rich ice-rafted debris (IRD) samples in cores LV90-8-1 and LV90-9-1 (Fig. S7). Grains larger than 250 μm account for 5% to 42% (~12% on average) in the sand fraction of IRD samples (Figs. S5 and S7), but less than 2% in almost all surface samples, except a near shore sample LV77-16. Grain size data of surface samples are from Ref. 81.

with the age pattern observed in the > 63 μm zircon fraction (Fig. 4). This suggests that some grains within the sand-rich IRD were transported by icebergs from East Siberia in the late or post-glacial stages, likely within the past three glacial cycles (Figs. 3 and 4). Icebergs form during calving from an ice shelf or ice cliff—that is, the edge of ice sheets or glaciers that extend into the ocean[50]. This implies that the East Siberia region was at least once glaciated[19,30], making an ice-free scenario unlikely.

However, the glaciation history of the East Siberia remains complex and highly controversial for decades, largely due to uncertainties in fragmented evidence and chronology. Three possible ice sheet configurations have been proposed. One suggests the formation of an ice sheet on the East Siberian continent[51], with an ice shelf extending over the East Siberian continent and into the shelf. The second scenario proposes an ice sheet that covered the East Siberian continental shelf[19], extending across the northern East Siberian Sea, northern Chukchi Sea, and Chukchi Borderland (Fig. 1), with a grounding line roughly at depths of ~300–1200 m (Supplementary Fig. S12). A third possibility involves a massive ice shelf covering the central Arctic Ocean, ~1000 m thick, which may have been a floating extension of an East Siberian ice sheet[30]. Some evidence, such as the ages of the oldest sediments covering glacial deposits or landforms[30,52–54] and cosmogenic exposure ages of bedrock on shelf islands[55,56], suggests that the East Siberian region may have been glaciated during MIS 6. Recent provenance studies also support the existence of the East Siberian Ice Sheet[6,57,58]. Here, by utilizing a distinct IRD provenance indicator from East Siberia and accounting for chronology uncertainties, our study demonstrates that East Siberian continent or continental shelf underwent multiple glaciations during past glacial-interglacial cycles. According to the available chronology, it is likely that the East Siberian experienced at least 2 to 3 glaciations during the last glacial-interglacial cycle (from MIS 5 to MIS 2, Figs. 3 and 4). As previous modelling studies have shown, once an ice sheet forms over the East Siberian regions, it can trigger regional warming feedback that accelerates its melting, leading to rapid fluctuations in ice sheet extent[59,60].

In summary, the coarse sand and zircon grains, along with the subglacial microfeatures of quartz grains from the sand-rich IRD

sediments in the central Arctic Ocean, provide strong evidence of iceberg transportation. The zircon age distribution, with a pronounced peak around 90-110 Ma, serves as a distinctive provenance indicator for IRD sourced from the repeatedly glaciated landscapes of East Siberia. The waxing and waning of the East Siberian Ice Sheet should have profoundly influenced both the polar and global climate[61]. Freshwater input from melting ice masses at the end of a glacial period would have affected the Atlantic meridional overturning circulation[62] and global sea levels[60]. Our provenance indicator can be applied to older Arctic sediments, facilitating a more comprehensive reconstruction of the East Siberian Ice Sheet[20,59]. Future research should aim to refine the extent of ice coverage in East Siberia and reassess its implications for climate dynamics, as well as its potential influence on human dispersals via the Bering Strait[63].

## Methods

### Sampling strategy and core logging
Surface sediments from Eurasian continental shelves and the two adjacent gravity cores LV90-8-1 and LV90-9-1 from the southern Makarov Basin (Fig. 1) were collected during the China-Russian Joint Arctic Expeditions and the 9th Chinese National Arctic Research Expedition (see detailed sampling information in Supplementary Table S1). Photos, bulk density, and color reflectance (Fig. 3) of the archive half of cores were obtained at a 1 cm resolution using the Geotek Multi-Sensor Core Logger by Ref. 29. X-ray radiograph was obtained using an Itrax XRF core scanner at 0.5-cm resolution. The two cores are well correlated lithostratigraphically based on bulk density and color reflectance (Fig. 3). Samples for detrital zircon U-Pb dating, grain-size analysis and quartz surface microfeature analysis were collected from the working half of LV90-8-1 and the lower LV90-9-1 (GP 6) (Fig. 3).

### Chronology
The chronologies of cores LV90-8-1 and LV90-9-1 are based on AMS $^{14}$C dating of total organic carbon (TOC)[29] and lithostratigraphic correlations to adjacent well-dated cores (PS2757-8 and 29-GC1) on the Lomonosov Ridge (Fig. 3). As the uncalibrated AMS $^{14}$C ages of TOC are older than the actual age of deposition due to the presence of old carbon, which was transported by ocean currents, sea ice or iceberg

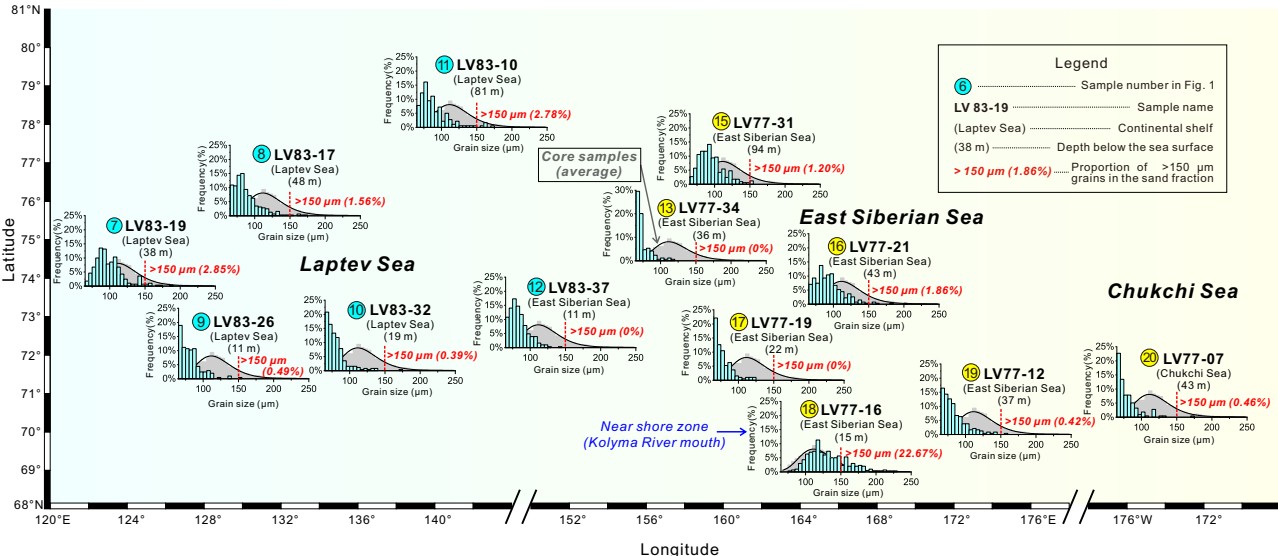

**Fig. 6 | Grain size distribution of 63–250 μm zircon grains from the surface samples on the Laptev Sea, East Siberian Sea and Chukchi Sea.** The vertical scale shows frequency per 5 μm bin normalized to total sand fraction (> 63 μm). The gray histograms in the background of each sample represent the zircon grain size distribution from all sand-rich ice-rafted debris (IRD) samples in cores LV90-8-1 and LV90-9-1. Note that zircon grains larger than 150 μm account for ~6 % to ~18% (~13% on average) in the sand fraction of IRD samples (Fig. S7), but less than 3% in almost all surface samples, except a near shore sample LV77-16. Source data are provided in Supplementary Data S2.

from the continent or continental shelf[54,65], we use these TOC [14]C ages to constrain the maximum depositional ages.

LV90-8-1 and LV90-9-1 are well correlated to sediment cores PS2757-8 and 29-GC1 using bulk density and sediment color (Fig. 3). However, there are two proposed age models for cores PS2757-8 and 29-GC1. The first age model was established using various dating techniques, including paleontological, magnetostratigraphic, and optically stimulated luminescence (OSL) dating (Fig. 3)[30,31,66]. The identification of MIS 3 and MIS 1 in core PS2757-8 was based on occurrence of dinocyst *Operculodinium centrocarpum*[67]. The base of PS2757-8 has previously been assigned an MIS 6 age through correlation to well-dated records on the Laptev and Barents Sea slope[68], and this age assignment was further supported by recent rock magnetic data and OSL dating in adjacent core 29-GC1[66]. The second age model, proposed by recent studies, uses excess U-series isotopes ($^{230}$Th$_{xs}$, $^{231}$Pa$_{xs}$)[32,33] to have identified a "$^{230}$Th$_{xs}$ extinction age" (~ 231 ka, early MIS 7) at a depth of ~5.9 ± 1.5 m in core PS2757-8 (Fig. 3), and dates the base of core to MIS 8.

Both age models are presented here (Fig. 3), as each has its own limitations. The first model, which lacks absolute ages older than MIS 3, incorporates OSL ages[66] but may underestimate ages for sample older than 70 ka (residual dose > 200 Gy) due to the natural saturation of the OSL signal in quartz[69]. The second model, based on $^{230}$Th$_{xs}$, showing strong variability down-core, possibly influenced by sea ice[32]. Furthermore, Purcell et al. (33) highlighted the large uncertainty of "$^{230}$Th$_{xs}$ extinction age" in core PS2757-8 compared to sites with lower sedimentation rates. Despite these uncertainties, both models support the conclusion that the East Siberian continental shelf has undergone glaciation during multiple glacial periods, likely over the past three glacial-interglacial cycles.

The boundaries between glacial and interglacial periods in cores LV90-8-1 and LV90-9-1 were determined based on abrupt changes in a* values. Two points showing the greatest variation in a* values at the transition between glacial and interglacial periods were identified (Fig. 3), and the midpoint of the interval between these points was defined as the boundary. Since no distinct abrupt a* change was observed between MIS 3 and MIS 2, the boundary was estimated using AMS [14]C dating results.

## Detrital zircon U-Pb dating

~300 g bulk sediments were used for extracting zircon grains, following conventional mineral separation techniques[21]. Heavy minerals in the samples were first extracted by water rinsing, followed by magnetic separation to extract nonmagnetic minerals. The nonmagnetic minerals were wet-sieved through a 63-micron sieve to extract the sandy components. Zircon grains were handpicked from the sandy nonmagnetic heavy minerals under a microscope, then mounted in epoxy resin and polished for dating. U-Pb isotopes of zircons were measured using an Agilent 7700x ICP-MS with a New Wave 193 nm laser ablation system at the Laboratory of Earth Surface Process and Environment, Nanjing University. The laser beam diameter was 37 μm with a 6 Hz repetition rate. We chose the outermost edges of zircon grains as the laser ablation points and avoided the xenocrystic core, voids and inclusions, based on the cathodoluminescence (CL) images (Supplementary Fig. S13) of zircon grains. Zircon 91500 was used as an external standard for isotopic fractionation correction[70], and GJ-1 zircon was used as a secondary reference material to monitor instrumental reproducibility and stability[71]. NIST 610 was used as the standard for normalizing the unknown U, Th, and Pb contents[72]. Glitter 4.4.2 was used to process the raw ICP-MS data[73]. Common Pb was corrected following the method of Ref. 74. Zircon particle grain ages were obtained using the following protocol: 1) Individual zircon discordance values < 10%, in agreement with the concordant ages of published data from potential sources; 2) $^{206}$Pb/$^{238}$U ages were used for zircon grains with ages <1000 Ma and $^{207}$Pb/$^{206}$Pb ages for older grains. 3) Intercept ages were used if the common Pb correction succeeds for discordant zircons. All zircon U-Pb ages and 1σ uncertainties are reported in Supplementary data S2.

## Age peak identification in zircon U-Pb age distribution

Zircon U-Pb age distribution are illustrated by kernel density plots (KDE) and frequency histograms (Figs. 2 and 4, Supplementary Fig. S11). KDE is a non-parametric statistical method that estimates the probability density function of zircon U-Pb ages[75], which quantitatively identifies age peaks. It works by placing a kernel function, typically a Gaussian (normal) function, at each measured zircon

U-Pb age and then summing these kernels at each position along the x-axis to create a smooth curve representing the density distribution.

The KDE at each position along the x-axis (x) is calculated using the following formula[75]:

$$f(x) = \frac{1}{nh} \sum_{i=1}^{n} K\left(\frac{x - x_i}{h}\right) \quad (1)$$

Where, $f(x)$ is the estimated density at each position along the x-axis, $n$ is the number of zircon U-Pb ages, $h$ is the bandwidth that controls the width of the kernel, $K$ is the kernel function, $x_i$ is the i-th zircon U-Pb age.

In practice, the kernel function $K$ is typically chosen to be a Gaussian, defined as:

$$K\left(\frac{x - x_i}{h}\right) = \frac{1}{\sqrt{2\pi}} \exp\left(-\frac{1}{2}\left(\frac{x - x_i}{h}\right)^2\right) \quad (2)$$

Then the age peaks (local maximum) were identified in the curve if a given $f(x_j)$ satisfies the condition: $f(x_j) > f(x_{j-1})$ and $f(x_j) > f(x_{j+1})$. Where $x_{j-1}$ and $x_{j+1}$ are the adjacent positions on either side of $x_j$ along the x-axis.

The bandwidth $h$ is a critical parameter that determines the smoothness of the estimated density function and the peak identification. The bandwidth determines the range of influence that the kernel density function of each zircon age has on the x-axis. A small value for $h$ results in a more jagged, less smooth estimate, while a large value for $h$ leads to over-smoothing. Here, we apply a consistent bandwidth, 15 Myr, to both surface and core samples. This allows for a direct comparison of the age peaks between the core and the surface samples. The choice of 15 Myr as the bandwidth is based on the minimal age difference of 30 Myr between the peaks at 90–110 Ma and 140–160 Ma. A bandwidth larger than 15 Myr would cause the kernel density functions of these two age groups to overlap, making it difficult to distinguish between them. On the other hand, a smaller bandwidth would make the curve less smooth and potentially distort the peak shapes. Thus, 15 Myr is selected to balance smoothness and peak resolution effectively.

Additionally, the sample size of zircon ages influences peak identification, with smaller sample sizes potentially overlooking minor peaks[76,77]. In our samples, the 90–110 Ma peak accounts for 2–8% of the age distribution (Supplementary Table S2). Statistical analysis and numerical simulations of zircon dating show that for sample sizes exceeding 500 grains, the likelihood of missing a 2% peak is negligible[78]. Even with a smaller sample size of 100 zircon grains, the 90–110 Ma peak is still clearly identifiable, with the likelihood of missing a 2% peak around 13% and a 5% peak about 5%[78]. In our case, the zircon age distributions from >63 μm grains exceed 500 grains in sample size (except for mid-GP6) (Fig. 4), while those from > 150 μm grains are around 100 (Supplementary Fig. S11). Therefore, the zircon grains dated in our samples reliably capture the 90–110 Ma peak, regardless of the sample size.

The source codes that plot KDE and identify age peaks can be found in Supplementary Text S5.

## Grain size analysis of bulk sample
The grain size analysis of sediment from cores LV90-8-1 and LV90-9-1 was performed at 1 cm resolution. Prior to instrument measurements, about 1 g of sample was dissolved with 15 ml of 15% hydrogen peroxide and treated at 20 °C for 24 h, followed by a 2-h treatment in a water bath at 85 °C. This process was repeated three times to eliminate all organic matter. Next, 5 ml of 3 mol/L hydrochloric acid was added at 20 °C for 24 h to remove the carbonate fraction. The residual samples were then rinsed with distilled water three times. Subsequently, 20 ml

of 2 mol/L sodium carbonate solution was added, and the samples were treated at 85 °C for 4 h in a water bath to remove the biogenic silica fraction. After rinsing with distilled water, the residual materials underwent a 1-min treatment in an ultrasonic cleaner to disaggregate potential sample aggregates. The grain size measurement was conducted in the laboratory of the First Institute of Oceanography, using a Mastersizer 3000 laser diffraction particle size analyzer. To ensure repeatability, each measurement was performed twice. All grain size data are reported in Supplementary data S3.

## Grain size analysis of zircon grains
The grain size of detrital zircon samples from the surface and cores LV90-8-1 and LV90-9-1 was measured using the image processing software ImageJ. Transmitted light images of zircon grains served as the base in ImageJ, as they clearly displayed the grain boundaries. Prior to imaging, the scale was calibrated. The transmitted light images were converted into binary black-and-white images in ImageJ to define the zircon grain boundaries. The Feret diameter, representing the maximum dimension of each zircon grain, was then calculated as the grain size. The raw data of zircon grain size are reported in Supplementary data S2.

## Quartz surface microfeature analysis
Six samples were collected from cores LV90-8-1 and LV90-9-1 for quartz surface microfeature analysis (see sample name and locations in Supplementary Fig. S9). 10 g of bulk sample were dissolved in 0.5 mol/L acetic acid overnight to remove the carbonate component, followed by treatment with hydrogen peroxide to remove the organic matter. The residual materials were then ultrasonicated for 30 min in a dispersant solution (0.05 mol/L sodium hexametaphosphate) to peel off the fine particles attached to the large particles. Afterwards, the samples were wet-sieved to extract components larger than 63 μm. The sand grains were pasted on the conductive adhesive after drying and painted with gold powder. Surface microfeatures were analyzed using a LEO 1430VP scanning electron microscope (SEM). Energy dispersive spectroscopy (EDS) was used to verify that the particles were quartz. Twenty images at 400X resolution were captured, and microfeatures of more than 100 coarse quartz grains (mostly > 100 μm) were randomly counted for each sample. The counting was performed by a single individual. Three replicate subsamples, each containing 30–40 quartz grains, were analyzed to improve reliability. Sample name was unknown to the operator to avoid subconscious bias. The surface microfeature classification and interpretation model applied in this study (Supplementary Figs. S8 and S9; see details in Supplementary Text S4) follows the Ref. 45. The counting data are reported in Supplementary data S4.

## Mass accumulation rates of ice-rafted debris
Although IRD in this study is defined as particles larger than 63 μm, we calculate the mass accumulation rates (MARs) of terrigenous grains larger than 150 μm as ">150 μm IRD", to ensure comparability with data from nearby cores like IODP Site 302[34].

MARs of >150 μm IRD were calculated using the following formula[34]:

$$\text{IRD MARs} = CS\% \times IRD\% \times DBD \times LSR \quad (3)$$

Where CS% represents the percentage of particles larger than 150 μm. IRD% is the proportion of terrigenous grains within this size fraction. DBD denotes the dry bulk density. LSR is the linear sedimentation rate.

The CS% values were derived from grain size data obtained in this study. Although microscopic counting was not used to quantify the proportion of terrigenous grains in the sandy fraction, EDS data, collected during the quartz surface morphology analysis, showed that terrigenous minerals (e.g., quartz, feldspar, etc.) accounted for over

95% of the grains in sand-rich layers. This percentage was used to approximate IRD%. DBD was calculated by multiplying the wet bulk density[29] by the sediment water content. Water content was determined by collecting sediment samples at 10 cm intervals using syringes prior to subsampling. The samples were weighed before and after oven-drying to calculate water content (Supplementary Data S1). The LSR was derived from the first age model[30,31], which is adopted for IODP Hole 302[34]. The calculated IRD MAR values are presented in Figs. S5e and S5f.

## Data availability
All data are available in the main text or the supplementary information.

## Code availability
All codes are available in the supplementary information.

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

## Acknowledgements

We are grateful to all crew and captains of vessels for sample collection during cruises. We thank Dr. Zhi Dong for helpful discussion on the manuscript. This work was supported financially by the National Natural Science Foundation of China (42130412) to X.S. (Shi), the National Key Research and Development Program of China (2023YFF0804600) to Z.Y., the National Natural Science Foundation of China (42206075) to H. F., the Marine S&T Fund of Shandong Province for Pilot National Laboratory for Marine Science and Technology (Qingdao) (2018SDKJ0104-3) to X.S. (Shi), the Taishan Scholar Program of Shandong (tspd 20181216) to X.S. (Shi), the China Postdoctoral Science Foundation (2022M723710) to H. F., the Russian state budget (121021700342-9) to Y. V., A. A., A. B., and the Russian Scientific Foundation (22-17-00118) to Y. V., A. B.

## Author contributions

Conceptualization: X.S. (Shi), Z.Y., Z.Z.; Methodology: H.F., H.Z., X.S. (Shan), G.Y.; Investigation: Y.L., L.H., Y.V., A.A., A.B., H.L.; Visualization: H.F.; Funding acquisition: X.S. (Shi), H.F., Z.Y., Y.V., A.A., A.B.; Project administration: X.S. (Shi); Supervision: X.S. (Shi), Z.Y.; Writing–original draft: H.F., Z.Y., Z.Z.; Writing–review & editing: X.S. (Shi), H.L., H.Z., Y.L., X.S. (Shan), J.D., L.D., Y.V., A.A.

## Competing interests

Authors declare that they have no competing interests.
