## [Transparent Peer Review file · Nature Communications]

Arctic zircon U-Pb ages reveal multiphase glaciations in East Siberia during the late Quaternary

Corresponding Author: Professor Xuefa Shi

A version of this paper was originally rejected for publication by Nature Communications, however that decision was reconsidered after appeal by the authors.

Version 0:

Reviewer comments:

Reviewer #1

(Remarks to the Author)

Overview: The authors have generated new data that will be very useful to the scientific community and provide a good foundation for future work on the history of glaciation in the north polar region. The paper does not propose a new idea, but rather new evidence to complement previous interpretations on the controversial topic of the presence of an East Siberian ice sheet. The deep-sea cores described here appear to be excellent archives of multiple glacial cycles and the detrital zircon provenance data are sound, and interpretations are logical. However, I did not find the depositional process interpretation to be convincing. This is critical to the framing and significance of this paper and as such, I think substantial revision is needed prior to publication.

I provide some comments below along with many notes on the PDF text and figures.

Lines 138-141 and in the Discussion. The big picture 'story' of this paper hangs on the interpretation of the process by which sand was deposited. However, the interpretation isn't well supported with the current text and requires more substantiation. For example, what is the explanation for the nearly complete absence of coarse sand, granules or pebbles in the 'IRD layers'? IODP site 302, near the cores, contains pebbles and granules along with in IRD (St. John, 2008) – why would core 8-1 and 9-1 lack this coarse component unless they were deposited by the same process? They also point out that coarse grains do not necessarily represent deposition by ice rafting. All-in-all, the uniformity of the 'IRD' sizes and lack of concentrated IRD-rich layers seems atypical and requires further examination/explanation. Could changes to bottom current strength have created this grain size distribution? There is tangential reference to this, but the arguments for/against this interpretation need to be discussed more clearly and thoroughly.

The discussion of quartz micromorphology throughout the paper is missing relevant literature (e.g., several papers by Immonen; Woronko (2016, Sed Geo); Mahaney (2002, Oxford Univ Press); Sweet and Soreghan (2010, JSR), etc.). Also, were replicates of any sample analyzed to evaluate reliability? Did one person do this work or multiple people? This work is highly dependent on the individual and the consistency issue needs to be addressed.

It is puzzling that till was not mentioned as a type of sediment on the continental shelf – has it been described? If so, it would have been illustrative to characterize quartz from till with SEM for direct comparison.

Lines 204-210. The text here includes many unstated assumptions and/or incomplete explanations. -Sediment in ice shelves typically melts out within a short distance of the grounding. Is there any evidence of a past grounding line on the shelf?

-What are the assumptions being made about the nature of ice sheet that may have occupied the continental shelf? Is there any evidence that it was cold based or warm based? How would this have affected sediment erosion, entrainment and deposition and how might that be reflected in the cores studied?

-What is meant by "an ice sheet or glaciers froze the East Siberian continental shelf"? I interpreted this to mean that ice was

frozen to the bed, but if that was the case, then erosion and comminution was likely to be minimal. This is inconsistent with what is presented elsewhere about the glacial environment.

Figure 2.

Displaying the data grouped by region doesn't show the significant variability within each group. This is important to be clear about and should be shown in the main body of the paper. Display all individual samples along with the aggregate samples on a multidimensional scaling plot to clarify the amount of variability between samples.

Figure 3. The interglacials typically have a small amount of sand (IRD) but glacial periods lack IRD until the very end. This pattern isn't mentioned in the text. What is the explanation for this pattern? See other comments on the figure.

Figure S3. What is the (n=X/Y) on the figure?

Figure 4. Visually and from the zircon proportion % on the right, B, D, G, and J look similar to each other, not just D and G. Additional statistical analyses are needed to better differentiate sample provenance and make the case for variable provenance.

Chronology - I did not find a table detailing the radiocarbon dating results. This needs to be published with the ages shown.

I appreciate this study represents resource intensive work and look forward to seeing it published in the future.
Kathy Licht

Reviewer #2

(Remarks to the Author)

This paper presents interesting results from detrital zircon U-Pb dating used to find provenances of sediments in two marine cores from the Makarov Basin, central Arctic Ocean. The linked provenances are in turn used to interpret the Arctic Ocean glacial history; more specifically the authors propose specifically that zircon dates in samples giving two peaks, ~90-100 Ma and ~150 Ma, are linked to an East Siberian source. Four such intervals are found in the sediment cores, and those are linked to four distinct glacial periods. The authors go on and suggest that this indicate that "the East Siberian continental shelf was occupied by an ice sheet or glaciers during multiple glacial periods."

The existence of an East Siberian ice sheet has been raised by several authors, and some evidences do exist, but the topic is far from resolved implying that it is very much welcomed with new views. The paper is well written and figures are perhaps a bit colorful, but clear. The dating methodology seems solid and the number of ages is impressive.

However, I do have some overall concerns that to my mind should be addressed.

First, it is a bit difficult to evaluate how robust the provenance linkages are: How significant are the zircon age peaks used as fingerprints of provenances?

This is hard to judge as Figure 2 has different scales on the y-axis, ranging from 0-20 to 0-80. For example, the peak for 149 Ma in F is represented by 6 dated grains, while this would not show up as a peak in for example in B where there are more than 6 grains dated to this age as far as I can judge from the graph, which in B has a y-axis scale of 0-80 instead of 0-20 as in F.

I would suggest here that a statistical cross-correlation between the records is used to really show that there indeed is a robust correlation between the peaks in Figure 2. Some of the peaks are obvious and clear, but some are not, simply as they are not comprised of that many dated zircons. I would not have suggested this if all peaks stood out clearly, and the fact that the peaks here are used to draw rather large conclusions on the glacial history. So, if we could be convinced, the story would to my mind be much stronger and one could leave this behind and go into details.

Second, the authors over-simplify results from previously published studies in an attempt to make their new results stand out as striking, contradictive and, perhaps, provocative. This is not necessary, instead it would be better to embrace the earlier studies and build on them accurately. They state, for example, that the "evolution of Northern Hemisphere ice sheet was more complex than previously thought". I'm not sure I agree that the evolution is expressed to be particularly simple in the majority of published papers; a potential ice sheet on the East Siberian Shelf has been raised in several works, some of which the authors cite, but many more in addition. For example, it can be found in earlier papers such as in Hughes et al. (1977, *Nature*), and papers by Russian geographer Grosswald. Furthermore, they state that their results challenge the notion of a "one-time glaciation in East Siberia occurred probably during MIS 6" and refer to Jakobsson et al. (2016, *Nature comm.*). This paper is about a floating ice shelf in the central Arctic Ocean, inferred from ice grounding at depths up to 1000 m. It is correct that this paper points to glaciation of the East Siberian Shelf as an important component for the ice shelf and should be cited as such, but it does not say that it only ever occurred during MIS 6. It even ends with "Finally, we emphasize that our results do not exclude the possibility that large ice shelves developed in the central Arctic Ocean during other glaciations than MIS 6. On the contrary, it seems likely that ice shelves were reoccurring components in most of the Quaternary glaciations considering the land-locked nature of the Arctic Ocean." In other words, it is the thickness and extent of the ice shelf that seems to be larger in MIS 6 (assuming the age model in their paper). Several more papers raise that ice shelves may have existed along the East Siberian Shelf, see for example Ye et al. (2020, <https://doi.org/10.1016/j.margeo.2020.106289>).

I think it is important to refer to previous results accurately, otherwise the scientific history will be skewed. I kind of got stuck on the two main points raised below and did not go much into details of the paper.

Reviewer #3

(Remarks to the Author)

Han Feng et al. have amassed a large number of U-Pb zircon dates from several surface samples from the East Siberian Arctic and sediment cores in the Arctic Ocean and used these for provenance of ice-rafted detritus (IRD). They use the >63 micron fraction to indicate iceberg IRD rather than sea ice IRD. Unfortunately, they did not take note of more recent papers that show IRD found in modern sea ice samples contain abundant sand greater than 63 microns (Darby, 2003; Darby et al. 2012, 2011; Reimnitz et al. 1998). In fact, a large number of ice floes containing sand were deemed to be anchor ice and these can contain even pebbles. They also use SEM texture on quartz grains to indicate glacial iceberg IRD rather than sea ice IRD. However, IRD in Arctic cores can be both sea ice and iceberg transported in the Pleistocene. There is no method to determine that only icebergs transported IRD to the central Arctic in the past. So even the presence of glacial features on quartz grains does not prove that only icebergs were the mode of transport because core samples show subrounded features as well as angular features. Furthermore, the general lack of nearshore and wave activity in the Arctic is not conducive to well rounded grains. Thus the basic premise of the Han Feng et al. paper that the core sediment sampled is iceberg derived is negated.

Secondly, They have no samples from the Axel Heiberg Island and Ellesmere Island and Banks Island areas that were major outlets for North American icebergs from the Laurentide and Innuitian Ice Sheets (Darby and Zimmerman, 2008). The only samples from the Axel Heiberg and Ellesmere Island areas were from previous studies using Sr - Nd provenance and these have a large overlap with other areas in the Arctic (see Supplemental Materials in this ms).

Finally, total IRD found in Fram Strait and cores near Denmark Strait (Darby and Zimmerman, 2008; Darby et al., 2017) show very little from the East Siberian Sea using a newer matching method for Fe Oxide fingerprinting that is overlooked by the authors of this submitted paper (Darby et al., 2015). This provenance Fe Oxide fingerprinting method is shown to be accurate to less than 2% error in source determination based on nearly 40,000 grain analysis using micro probe analysis of Fe grains for 14 elements (Darby et al., 2015). Also it is true as pointed out by the authors that heavy mineralogy is not the most effective provenance tool in the Arctic, but the mineralogy of hundreds of circum-Arctic samples has indicated that there are differences between general areas in the Arctic with regards to feldspar type and other light minerals (Myers and Darby, 2022). The authors need to revise their conclusions and make a better argument for iceberg IRD, perhaps by showing that the zircon grains are much larger than 250 microns.

Darby, D. A., Andrews, J. T., Belt, S. T., Jennings, A. E., and Cabedo-Sanz, P., 2017. Holocene cyclic records of ice-rafted debris and sea ice variations on the East Greenland and Northwest Iceland margins. *Arctic, Antarctic, and Alpine Research*, v. 49, no. 4, 649–672

Darby, D. A., Myers, W., Herman, S., and Nicholson, B., 2015, Chemical fingerprinting, a precise and efficient method to determine the source of sediments. *Jour. Sed. Res.*, v. 85, 247–253

Darby, D. A., 2014, Ephemeral formation of perennial sea-ice in the Arctic Ocean during the middle Eocene. *Nature Geoscience*, v. 7, p. 210-213

Darby, D. A., W. B. Myers, M. Jakobsson, and I. Rigor, 2011, Modern dirty sea ice characteristics and sources: The role of anchor ice, *J. Geophys. Res.*, 116, C09008

Darby, D. A., 2003, Sources of sediment found in sea ice from the western Arctic Ocean, new insights into processes of entrainment and drift patterns. *J. of Geophys. Res.*, v. 108, C8, 3257

Darby, D.A. and P. Zimmerman, 2008, Ice-Rafted Detritus Events in the Arctic During the Last Glacial Interval and the Timing of the Innuitian and Laurentide Ice Sheet Calving Events, *Polar Research* 27: 114-127

Myers, W. and Darby, D. A., 2022. A compilation of the silt and clay mineralogy from coastal and shelf regions of the Arctic Ocean. *Marine Geology*. P. 106948.

Reimnitz, E., McCormick, M., Bischof, J., Darby, D.A., 1998. Comparing sea-ice sediment load with Beaufort Sea shelf deposits: is entrainment selective? *Journal of Sedimentary Research*, 68:777-787

Version 1:

Reviewer comments:

Reviewer #1

(Remarks to the Author)

The authors have made substantial revisions to this manuscript and compared to the earlier submission, this paper makes a clearer case for coarser layers of sediment from deep sea cores being of ice-rafted origin. I have some additional comments that I'd like to see addressed. With those addressed, I think this work will make a valuable contribution to the discussion of an East Siberian ice sheet.

•In Fig. 1, it is hard to see the distribution of rocks that are considered sources to the offshore record. The magmatic rocks are highlighted, but they are not the only ones that could be contributing zircons. There seems to be a wide distribution of sedimentary rocks in this region through which the zircons may be recycled. State whether there are DZ data from extensive Neogene sedimentary rocks or not, and if so whether they contain grains 90-110 Ma. Also, the legend lists 'geological units', but time periods are listed, not units. Rocks shown by time period are not particularly helpful without additional information. Reconsider what kind of basemap may be more useful. Zircon-bearing rocks along with the age population in them would be best, but creating such a map would be a whole project by itself. This geographic region isn't my area of expertise, so I'm not sure what may be available to use as an alternative. The lower left 1/3 of the map isn't really needed, so it could be cut off so the remainder could be enlarged.

There is a typo on the map = North Ameriaca should be fixed

•Move Figure 2 to the supplement.

•The chronological control on the timing of sediments is not strong. Only one of the two cores was dated, and that chronology has numerous reversals. I think it would be helpful to readers to move Fig. S4 (+/- the log for PS2757-8) in place of Fig. 3 in the main text. The points with the dates shown on cores 9-1 and 29-GC1 should be in the main part of the paper.

•Fig S5. It is unusual that with all the granules/pebbles shown in A and E (in particular) there is nothing in the core photos indicating their presence. If there are x-radiograph images showing the core with the larger size fractions in situ, they should be included.

•Discussion

o The data in the right column of Figure 4 would be better in the supplement and replaced with Fig. S7. Based on results in Fig. S7, I recommend the authors add more nuance to their interpretations of the implications of this work. Samples 4-2 and 4-1 are convincingly like East Siberia sources, samples 2-1 and 5-1 are not, and the remainder seem to indicate varying relative inputs from East Siberia and Central Eurasia. Three possible ice sheet configurations, which configurations could explain the variations in input/relative strength of an East Siberian source?

o Clarify language around ice shelves. Icebergs need not be sourced from an ice shelf, would a tidewater margin have been possible in this region? We know from Antarctica that large ice shelves usually yield ice bergs with minimal entrained sediment as most melts out near the grounding line. Is there geomorphological evidence for grounding zone wedges on the East Siberian shelf that should be brought into this discussion?

Reviewer #2

(Remarks to the Author)

Dear Authors,

I find that that you have done a thorough revision of the manuscript. I have only a few minor comments to be considered below.

Line 57: Mixture of terms. Eurasian Basin includes Amundsen and Nansen basins, while Makarov Basin is a subset of Amerasian Basin. I suggest stating Amundsen Basin and Nansen Basin instead of Eurasian Basin, to be consistent with the nomenclature use for the Amerasian side. I propose also changing the map to reflect this.

Line 65: Note sure if reference 20 should be used, it is a paper that did not reach full publication, and remained in discussion. Why not include some older reference where it really began to be discussed:

Hughes, T.J., Denton, G.H., Grosswald, M.G., 1977. Was there a late-Würm Arctic ice sheet? *Nature* 266, 596–602. <https://doi.org/10.1038/266596a0>

Line 122: Consider writing "...interpreted to represent an oxidizing environment with reduced sea ice during interglacial periods." I would not go so far as to say it signifies seasonal ice only. There is no reference to back this up? The Mn cycles we first used as proxy of interglacial or interstadial conditions in: Jakobsson, M., Lovlie, R., Al-Hanbali, H., Arnold, E., Backman, J., Mörrh, M., 2000. Manganese and color cycles in Arctic Ocean sediments constrain Pleistocene chronology. *Geology* 28, 23–26. [https://doi.org/10.1130/0091-7613\(2000\)28<23:MACCIA>2.0.CO;2](https://doi.org/10.1130/0091-7613(2000)28<23:MACCIA>2.0.CO;2)

Line 131: "...we propose"... sounds like the two age models are proposed. Consider writing "we put forward two alternative..."

Line 186: Should be "sea-ice origin", not "sea ice-origin"

Line 2013. "Icebergs form during calving from an ice shelf..." An ice margin extending into the ocean does not always have a floating extension in the form of an ice shelf, it may also end in an ice cliff. So to be more accurate I suggest writing "Icebergs form during calving from an ice shelf or ice cliff - that is..."

Line 2017: I do not understand the use of the references here: "This implies that the East Siberia region was once glaciated, making an ice-free scenario unlikely^{18,30}".

If the references are moved to after ...glaciated, I believe it would be correct considering the content of the papers referred to: "This implies that the East Siberia region was once glaciated^{18,30}, making an ice-free scenario unlikely".

Line 225: I suggest a slight modification to make it accurate with respect to what was proposed in the paper referred to: "A third possibility involves a massive ice shelf, approximately 1000 m thick, that covered the entire Arctic Ocean, and which may have been a floating extension of an East Siberian ice sheet³⁰

Reviewer #3

(Remarks to the Author)

Review of Feng et al., resubmitted

This submission is well written, but it could be tightened up and shortened to make it more readable. It deals with an important issue, i.e., did a large ice sheet exist on the E Siberian Shelf during low sea level cold intervals as far back as MIS 6 if the stratigraphy is correct. The paper presents a large number of zircon dates that are valuable for future Arctic provenance studies.

There are very important assumptions made that are not stated and need to be addressed. Primarily, the question of whether the IRD and zircons found in the two cores studied are glacial or sea ice-rafted. Studies that show the importance of anchor ice are overlooked or mistaken for suspension freezing entrainment. The contention that any sea ice-rafted sediment to the core sites is entrained by suspension freezing is just not true. Previous studies show that extensive areas of sediment-laden sea ice in the central Arctic are anchor ice. These ice floes contain 100 to a thousand times more sediment than sea ice with sediment entrained by suspension freezing. Both the HOTRAX and LOMROG expeditions in 2005 and 2007 found 100's of kilometers of sea ice with high concentrations of sediment including large benthic shells and sediment coarser than 250 microns. Thus by down-playing sea ice-rafting, the paper is heavily biased toward iceberg transport.

Feng et al. point out that >250 micron quartz has some glacial features and that the texture of sediments on the Modern E Siberian Shelf are too fine-grained to supply zircons >150 or 250 microns. This is all true but it is based on the assumption of whether this shelf was covered by the same sediment that was fine-grained in texture during low sea level stands as it is today. This assumption needs to be pointed out and addressed. Surely, as sea level dropped and river gradients increased, coarser sediment would be transported to the shelf. This could extend to the glacial shoreline where this coarser sediment could be entrained by anchor ice. Another problem with the SEM quartz microfeatures of glacial origin is that there is no way to determine that the quartz grains with glacial features were derived from the E Siberian Shelf and not from sources like North America that were glaciated. Other studies clearly show that there are multiple sources in sediment cores in the central Arctic.

Because anchor ice entrains any sediment size on the seafloor, the extensive discussion of incipient movement of sand grains on the sea floor in the new Supplemental Materials section can be deleted. There is no need to suspend or move sediment for anchor ice to entrain it.

It is interesting that only total organic carbon is used for radiometric dates in the two studied cores. TOC dates have been shown to have large offsets with calcareous shell ¹⁴C dates (Hansen et al., 2022, Quat. Geochronology). Radiocarbon is good only back to about MIS 4 anyway, so not much help with MIS 6. Mn stratigraphy is fine but it has its issues with accuracy because minor excursions to cold conditions can deposit a grey layer mistaken for a full-blown glacial. However, despite these issues the stratigraphy of the two cores is about as good as it will ever get given the data at hand.

Specific lines that could improve the paper:

Line 46: ...icebergs calved from ice sheets as well as sediment entrained by anchor ice across the Arctic (8 + new reference)

Line 48: ...by ice sheets or sea ice.

Line 49-51: ...except for Fe grain fingerprinting...

Line 62: Fe grain fingerprinting is also very effective in distinguishing E Siberian sediment from other circum-Arctic sources.

Line 67: Zircon ages should be touted as an additional tool not an end-all method for provenance.

Line 92: The discussion of the Pechora River, Kara Sea etc zircon sources belongs in the Supplemental Materials. This entire paragraph could be eliminated and make the paper more readable and shorter. The following paragraph states that the zircon ages for the E Siberia area are unique and this suffices along with the existing Figure showing this. Besides it is already stated that the zircon sources for these other areas is discussed in Supplementary Materials.

Line 177: Zircon density is immaterial for ice-rafting as heavy grains are just as easily entrained by anchor ice as light grains.

Line 212: Iceberg transport is not the only plausible explanation but one of the most likely.

Line 226: Hopefully no one still proposes that a 1000m thick ice sheet covered the entire Arctic Ocean. But a third possibility instead of this absurd kilometer thick ice sheet would be that anchor ice entrained coarse sediment from the low-stand sea level of glacial conditions. It is very feasible that during dropping sea level river gradients increased and transported sand to the shoreline even to the shelf margin.

Version 2:

Reviewer comments:

Reviewer #1

(Remarks to the Author)

I have reviewed the revisions made by the authors and I am satisfied that they have addressed my comments. I appreciate the time they have taken to revise the figures and text to add more clarity and nuance.

Kathy Licht

Reviewer #2

(Remarks to the Author)

Dear Authors,

I commend you for patiently revising the manuscript carefully considering all comments. I do not have anything more to add, I think you have replied to all comments and addressed them properly.

Best Regards

Martin

Reviewer #3

(Remarks to the Author)

The authors have made a good faith effort to address all of my concerns and I have no further objection to publication.

**Reply to the reviewers’ comments on “Arctic zircon U-Pb ages reveal**
**multiphase glaciations in East Siberia during the late Quaternary” by Feng**
**et al.**

Content

Reviewer #1:2
 Overview.....2
 Comment 1 on depositional process2
 Comment 2 on quartz micromorphology7
 Comment 3 on East Siberian Ice Sheet..... 11
 Comment 4 on figure 2 13
 Comment 5 on figure 3 19
 Comment 6 on figure 423
 Other comments24
Reviewer #2:29
 Overview.....29
 Comment 1 on identification of zircon age peak30
 Comment 2 on ESIS evolution:36
Reviewer #3:39
 Comment 1 on differentiation of iceberg and sea ice origin:.....39
 Comment 2 on source region:46
 Comment 3 on provenance indicator of minerals49
References.....52

**Reviewer #1:**

**Overview**

The authors have generated new data that will be very useful to the scientific community
and provide a good foundation for future work on the history of glaciation in the north polar
region. The paper does not propose a new idea, but rather new evidence to complement
previous interpretations on the controversial topic of the presence of an East Siberian ice sheet.
The deep-sea cores described here appear to be excellent archives of multiple glacial cycles
and the detrital zircon provenance data are sound, and interpretations are logical. However, I
did not find the depositional process interpretation to be convincing. This is critical to the
framing and significance of this paper and as such, I think substantial revision is needed prior
to publication.

**Reply:**

We sincerely appreciate your positive feedback that recognizing our study providing an
excellent archives and new evidence for multiple glacial cycles, as well as its potential to serve
as a foundation for future research on the history of glaciation in the north polar region. In light
of your comments, we have emphasized these points in both the introduction and conclusion
sections to highlight the significance of our study. In response to your concerns and suggestions,
as well as those of other reviewers, we have made substantial revisions to the manuscript,
particularly including additional experimental analyses of detrital zircon U-Pb ages to provide
a more comprehensive and detailed explanation of the depositional processes. For a full
account of these revisions, please refer to the reply to major comments 1 to 6.

**Comment 1 on depositional process**

Lines 138-141 and in the Discussion. The big picture ‘story’ of this paper hangs on the
interpretation of the process by which sand was deposited. However, the interpretation isn’t
well supported with the current text and requires more substantiation. For example, what is the
explanation for the nearly complete absence of coarse sand, granules or pebbles in the ‘IRD

layers’? IODP site 302, near the cores, contains pebbles and granules along with in IRD (St.
John, 2008) – why would core 8-1 and 9-1 lack this coarse component unless they were
deposited by the same process? They also point out that coarse grains do not necessarily
represent deposition by ice rafting. All-in-all, the uniformity of the ‘IRD’ sizes and lack of
concentrated IRD-rich layers seems atypical and requires further examination/explanation.
Could changes to bottom current strength have created this grain size distribution? There is
tangential reference to this, but the arguments for/against this interpretation need to be
discussed more clearly and thoroughly.

**Reply:**

Thank you for raising these important points regarding the depositional processes. We
have addressed these concerns in the revised manuscript with the following additions:

**(1) Granules and Pebbles:**

Granules and pebbles are actually also present in some sand-rich layers of our cores. We
show some images of pebbles and granules found in our samples in Fig. S5 below. Another
evidence is the presence of voids in the late GP 6 layer (as shown in Fig. S5G), which indicate
the presence of granules and pebbles that fell out during core splitting. Therefore, we have
added “In addition, isolated granules and pebbles are found in these sand-rich layers
(Supplementary Fig. S5), further suggesting that these sand particles are IRD.” in *Page 7, line*
*146-147*.

**Fig. S5. Core image and sand content of typical sand-rich layers in core LV90-8-1 (A-F)**

**and LV90-9-1 (G, H).** Sample positions of detrital zircon U-Pb dating are marked by

homogeneous composition with bioturbation (mottling); The gradational contacts between adjacent beds indicate

continuous deposition of sediments, in which sands are typical ice rafted debris (IRD); (D, F)

Thin lamination with normal-graded sequence and sharp base indicating turbidite deposition.

(G) Voids left by fallen granules and pebbles indicate IRD deposition. (H) Homogeneous

composition with no bioturbation or beddings. GP = Glacial period. IGP = Interglacial

period.

**(2) IRD MARs:**

However, as not all sand-rich layers in the IODP Site 302 cores contain granules or pebbles

(St. John, 2008), the presence of granules and pebbles is not a definitive criterion for identifying

IRD. Here, by comparing IRD fluxes, we demonstrate the consistency between the IRD in our

cores and those in the nearby IODP 302 core. We calculated the mass accumulation rates

(MARs) of $>150\ \mu\text{m}$ IRD in cores LV90-8-1 and LV90-9-1 (a new Fig. S6 has been added, see
 below), using the same methodology from St. John (2008) for IODP Site 302. The $>150\ \mu\text{m}$
 IRD MARs in the sand-rich layers range from 0.05 to 0.44 $\text{g}/\text{cm}^2/\text{kyr}$ (Fig. S6E,F), which falls
 within the range reported for IODP Site 302 (0.02–0.36 $\text{g}/\text{cm}^2/\text{kyr}$) and other Arctic cores (St.
 John, 2008 and references therein), indicating no order-of-magnitude differences between the
 two sites. This similarity suggests that the depositional processes for coarse sand components
 in our cores are comparable to those at Site 302.

This information has been added in *Page 7, Line 147-150*:

“The MARs (mass accumulation rates) of IRD in these sand-rich layers range from 0.05
 to 0.44 $\text{g}/\text{cm}^2/\text{kyr}$ (Supplementary Fig. S6), which is comparable to the range of IRD MARs
 (0.02–0.36) observed in the nearby IODP (Integrated Ocean Drilling Program) 302 core³⁴.”

**Fig. S6.** Content and mass accumulation rate of coarse sand grains in core LV90-8-1 and
 LV90-9-1. (A, B) Proportion of $>150\ \mu\text{m}$ and $>250\ \mu\text{m}$ grains in all particles. (C, D) Proportion
 of $>150\ \mu\text{m}$ and $>250\ \mu\text{m}$ grains in all sand particles ($>63\ \mu\text{m}$). (E, F) Mass accumulation rate
 (MARs) of $>150\ \mu\text{m}$ and $>250\ \mu\text{m}$ grains. The calculation process of MARs is detailed in the
 Materials and methods. Detrital zircon dating samples are marked in E and F, same as Fig. 3.

**(3) Bottom currents:**

Previous studies (e.g., McCave et al., 1995; McCave and Hall, 2006) have shown
that the relationship between the mean grain size of sortable silt (\overline{SS} , 10–63 μm) and its
proportion in the < 63 μm fraction (SS%) can indicate the influence of bottom currents.
However, in both sand-rich and non-sand-rich layers, the correlation between \overline{SS} and
SS% is weak ($r < 0.5$, slope < 0.2) (a new Fig. S11 has been added), which is far below
the standards suggested by McCave et al (2019) ($r > 0.7$, slope > 0.7), indicating
minimal influence of bottom currents. This is likely due to the low bottom current
velocities (< 5 cm/s) in the study area (Woodgate et al., 2001). Such low velocities are
insufficient to mobilize particles larger than 63 μm (Miller et al., 1977).

**Fig. S11. Correlation between the mean grain size of sortable silt (\overline{SS} , 10–63 μm)**
**and its proportion within the <63 μm fraction (SS%) in the sediments from LV90-**
**8-1. (A) Sediments from sand-rich IRD layers; (B) Sediments excluding the sand-rich**
**IRD layers. Original data can be found in Source Data S3.**

This information has been added in *Page 10, Line 206-211*:

“Additionally, to assess the role of bottom currents in transporting these particles, we examined the relationship between mean grain size and the percentage of sortable silt (10–63 μm)⁴⁶. The observed relationship for samples in the sand-rich IRD layers,

with an R value < 0.5 and a slope value < 0.2, does not support the influence of bottom
currents⁴⁶ (Supplementary Fig. S11). Moreover, the estimated bottom current velocity
at the core site (< 5 cm/s) is insufficient for transporting coarse particles⁴⁷.”

**(4) Iceberg-IRD vs. Sea-Ice-IRD:**

Once sand-rich layers are identified as IRD, distinguishing between iceberg and
sea-ice sources becomes crucial. In the revised version of the manuscript, we have
refined our argument to focus on zircons larger than 150 μm (an equivalent grain size
for 250 μm quartz) as evidence for iceberg transport. A detailed explanation can be
found in our response to Reviewer 3's first comment below. Based on this, we obtained
~2500 new zircon ages through additional experiments, rewrote the first two paragraphs
of the discussion section (please see *Page 8-9, line 167-184*), redrew Fig. 4, and added
new Fig. 5, Fig. 6 and Fig. S8.

**Comment 2 on quartz micromorphology**

(1) The discussion of quartz micromorphology throughout the paper is missing relevant
literature (e.g., several papers by Immonen; Woronko (2016, Sed Geo); Mahaney (2002,
Oxford Univ Press); Sweet and Soreghan (2010, JSR), etc.). Also, were replicates of
any sample analyzed to evaluate reliability? Did one person do this work or multiple
people? This work is highly dependent on the individual and the consistency issue needs
to be addressed.

(2) It is puzzling that till was not mentioned as a type of sediment on the continental
shelf – has it been described? If so, it would have been illustrative to characterize quartz
from till with SEM for direct comparison.

(3) Line 171: Two points here need to be addresses:

1) See Woronko, 2016

2) This depends on the (a) history of sediment prior to entrainment and (b)time/distance
spent in that depositional environment.

(4) Line 175: This has been documented in subglacial environments (e.g., Graly et al.,
2014, *Geology*) and should be considered.

**Reply:**

**(1) Method on quartz micromorphology:**

We have added the relevant literature in both the main text and the supplementary
materials (Supplementary Text S4). In the revised version, we have addressed the
reliability concern by analyzing three replicate subsamples, each containing 30–40
quartz grains, rather than analyzing a single batch of over 100 grains as in the previous
version. This follows the method described by Mahaney (2002). An updated Fig. S10
(see below) presents error bars for the replicate subsamples.

All analyses were conducted by a single individual. While single-person analysis
may introduce some degree of subjective bias, this does not affect our ability to
distinguish between sea-ice and ice-sheet sources. Culver et al. (1983) demonstrated
that although operator variance can be considerable in recognizing individual surface
features, it is negligible when differentiating samples from distinct depositional
environments. Our co-authors believe that involving multiple operators with varying
levels of training could introduce greater inconsistencies. This may explain why most
SEM-based microfeature analyses are rarely conducted by multiple analysts. We added
this information in Materials and methods section for quartz micromorphology in *Page*
*19, line 417-421*:

“The counting was performed by a single individual. Three replicate subsamples,
each containing 30–40 quartz grains, were analyzed to improve reliability. Sample
name was unknown to the operator to avoid subconscious bias. The surface
microfeature classification and interpretation model applied in this study
(Supplementary Figs. S9 and S10; see details in Supplementary Text S4) follows the
Ref. 43.”

Grain roundness
 Grain relief
 Mechanical features
 chemical features

Fig. S10. Microfeature percentage of quartz grains in sandy samples from

Holocene sediments (A) as well as sand-rich IRD layers (B-F) of core LV90-8-1 and

LV90-9-1. Error bars represent 1 SD (Standard Deviation), calculated from the analysis

of three subsamples. Features affected solely by glaciation (Subglacial features) are

framed with dotted lines. Note that quartzs in Holocene samples have fewer mechanical

features and subglacial features than those in sand-rich IRD samples.

**(2) Till's quartz micromorphology:**

Numerous moraine ridges along the shelf edge of the East Siberian continental
shelf have been identified through acoustic and seismic data in previous studies (e.g.
Jakobsson et al., 2016; Lehmann and Jokat, 2022; Shen et al., 2024. See a new Fig. S12
in Comment 3). Unfortunately, samples from these ridges are unavailable in this study.
In Supplementary Text S4, we have added descriptions of till characteristics from other
regions to demonstrate that the microfeatures of glacial till are consistent with those
observed in our sand-rich IRD samples. Notably, showing microfeatures of glacial till
can help us explain some previously puzzling questions, such as why subglacial
environments can also produce subrounded forms and heavy weathering. Please see
Supplementary Text S4 for details.

**(3) Subglacial vs. periglacial microfeatures:**

Woronko (2016) suggested that quartz grains influenced by subglacial and
periglacial environments share certain mechanical microfeatures, such as conchoidal
fractures, breakage blocks, and fracture planes. This raises the possibility that grains
with these features may not necessarily have experienced subglacial conditions during
glacial periods but could instead have been affected by periglacial processes and
subsequently transported to the deep sea by sea ice during postglacial sea-level rise.

To address this issue, we counted features solely associated with subglacial
environments, such as step-like fractures and subparallel linear fractures. These features
were identified in over 60% of the grains (as shown in Fig. S10 above), indicating that
the majority of the grains likely experienced subglacial conditions.

We add this information in *Page 9, line 189-195*:

“In contrast, quartz grains from sand-rich IRD samples are typically subangular
and display numerous mechanical microfeatures, including characteristic subglacial

microfeatures including step-like fractures and subparallel linear fractures^{42,43}
(Supplementary Fig. S10). More than 60% of the quartz in the IRD layer show these
subglacial microfeatures (Supplementary Fig. S10). While quartz grains in periglacial
environments can also exhibit mechanical features⁴², they lack these typical subglacial
microfeatures observed here. These subglacial microfeatures indicate a signature of
subglacial abrasion and crushing⁴⁰⁻⁴³.”

**(4) The preservation of microfeatures**

Although we have not found studies on whether long-distance iceberg transport
affects the preservation of microfeatures, other transport processes are known to modify
subglacial features. For example, Křížek et al. (2017) demonstrated that subglacial
features in till rapidly diminish after being transported by glacial meltwater. This
suggests that the subglacial features observed in our sediment samples have likely not
undergone significant fluvial reworking (see Supplementary Text S4).

**Comment 3 on East Siberian Ice Sheet**

Lines 204-210. The text here includes many unstated assumptions and/or incomplete
explanations.

(1) Sediment in ice shelves typically melts out within a short distance of the grounding.

Is there any evidence of a past grounding line on the shelf?

(2) What are the assumptions being made about the nature of ice sheet that may have
occupied the continental shelf? Is there any evidence that it was cold based or warm
based? How would this have affected sediment erosion, entrainment and deposition and
how might that be reflected in the cores studied?

(3) What is meant by ‘an ice sheet or glaciers froze the East Siberian continental shelf’?

I interpreted this to mean that ice was frozen to the bed, but if that was the case, then
erosion and comminution was likely to be minimal. This is inconsistent with what is
presented elsewhere about the glacial environment.

(4) Line 219: Is this supported by seismic data or sediments from the shelf indicating
ice sheet cover?

**Reply:**

In the revised version, we provided evidence and explanations regarding the nature
of the East Siberian Ice Sheet (ESIS), including:

**(1) Grounding lines and ESIS outline:**

The evidence for the grounding line primarily comes from two sources: the edge
of the seabed glacial landforms (like glacial lineation) and the location of the grounding
zone wedge (GZW) (See a new Fig. S12 below). The grounding line of the ESIS is
approximately at a depth of 300-1200 meters (Shen et al., 2024 and references therein),
and the extent of the ESIS is outlined accordingly (Niessen et al., 2013). However, the
location of our cores is still ~300 km away from the grounding line (Fig. S12), which
is beyond the influence of the GZW. Batchelor and Dowdeswell (2015) suggest that the
length of most GZW does not exceed 15 km. Therefore, our sediments are not derived
from rapid melting near the grounding line of the ice shelf, but rather from
disintegrating icebergs transported by the Transpolar Drift.

**Fig. S12. Glacial landforms identified on the East Siberian continental margin.** (A)
Grounding zone wedges (GZWs) within the De Long glacial trough⁵⁰; (B) GZWs and

Mega-scale Glacial Lineations (MSGs) on the East Siberian Sea continental margin¹⁸;
(C) MSGs on the Arliss Plateau¹⁸; (D) GZWs, MSGs, Morainic Ridges and ice-
originated plowmarks on the Chukchi Sea continental margin and Chukchi
Borderland^{17,18,30,118-121}. White dashed line shows the proposed extent of the East
Siberian Ice Sheet¹⁸.

We added the information of ESIS grounding line and its possible extent in *Page*
*10, line 223-225*.

“The second scenario proposes an ice sheet that covered the East Siberian
continental shelf¹⁸, extending across the northern East Siberian Sea, northern Chukchi
Sea, and Chukchi Borderland (Fig. 1), with a grounding line roughly at depths of ~300–
1200 m (Supplementary Fig. S12).”

**(2) Thermal regime**

The thermal regime of past glaciers is typically reconstructed through glacial
landforms (e.g., Sugden, 1977; Kleman and Glasser, 2007). For example, areas without
glacial traces are interpreted as having a frozen bed (cold-based glacier), while those
with glacial erosional features are interpreted as ice streams (warm-based glaciers).
Previously identified glacial features (Fig. S12), which are mostly found along the shelf
edge (Niessen et al., 2013), suggesting that the East Siberian Ice Shelf (ESIS) likely
experienced some basal sliding along the shelf edge, not frozen to the bed, at least not
at the end of glacial periods. We have replaced the term "froze" with "glaciated" or "ice-
covered" throughout the text to more accurately reflect the nature of the ESIS. However,
no glacial features were identified in the central parts of the ESIS, which may indicate
a frozen bed or the possibility that glacial features were covered by sediments.

**Comment 4 on figure 2**

(1) Displaying the data grouped by region doesn't show the significant variability
within each group. This is important to be clear about and should be shown in the main

body of the paper. Display all individual samples along with the aggregate samples on
a multidimensional scaling plot to clarify the amount of variability between samples.

(2) Line 106: LV77-12 and LV83-37 don't 'clearly' have the double peak. (LV77-26
has a higher % than LV83-37.) ARC9-C24 also has the double peak. There is a lot more
texture in this data than figure 2 suggests, so adding an MDS plot is needed.

**Reply:**

In response to this suggestion, we have revised Fig. 2 (see below) to display the
age distribution of all surface samples. The samples are now grouped into two panels
based on the presence of a 90-110 Ma peak (a specific marker for the East Siberian Ice
Sheet), instead of arbitrarily grouping them by region as in the previous version.

In Figure 2, we have now displayed more details in the data. For instance, the 150
302 Ma peak is absent or not clearly visible in certain East Siberian samples (e.g., LV77-07)
as you mentioned. As a result, we have revised the previous statement, changing the
"double peak of 90-110 Ma and 150 Ma" to the more convincing "single peak of 90-
110 Ma." as the fingerprint for the East Siberia.

**Fig. 2. Detrital zircon U-Pb age distributions of surface samples from circum-**
 **Arctic continental shelves.** Sample locations are shown in Figure 1. ~90-110 Ma,
 ~140-160 Ma, ~220–360 Ma, ~420–360 Ma, and ~1750–2000 Ma zircon age peaks
 are shaded. Each part of the circum-Arctic continental shelf has a distinct zircon age
 distribution. The western and central Eurasian samples (A-K) lack the ~90-110 Ma

peak which is prominent in the East Siberian samples (**M-U**). The Bering Sea sample
(**V**) has a unique ~60 Ma peak. The North American sample (**W**) is characterized by
1000–1500 Ma zircons that are absent from all the Eurasian samples (**A-U**). Hence,
the ~90-110 Ma age peak exclusively fingerprint the East Siberian provenance.

Thus, the description for Fig. 2 have been updated in *Page 5, line 89-107*.

“The zircon age distributions in these surface samples show regional distinctions
(Fig. 2), though they generally contain prominent zircon age peaks at ~220–360 Ma,
~420–560 Ma, ~1750–2000 Ma, and ~2500–3000 Ma (Fig. 2; the sources of these age
peaks are discussed in Supplementary Text S2)^{23,24}. In the Barents Sea, a small age peak
at ~185 Ma appears in the surface sediment near the Pechora River Mouth (Fig. 2A),
probably reflecting a source from the Ural Mountain formed in the continental collision
during the Early Jurassic²⁷. In the Kara Sea, two age peaks at ~140-160 Ma and ~180-
200 Ma (Figs. 2B-F) indicate the source from the Central Asian Orogenic Belt, located
in the upper reaches of the Yenisei and Ob rivers²⁸. In the Laptev Sea, the peak at ~140-
160 Ma (without the peak at 180–200 Ma) (Figs. 2G-K) is likely sourced from the
Kolyma-Omolon Magmatic Belt (~158–129 Ma) (Fig. 1)²⁹. On the East Siberian
continental shelf (including the East Siberian Sea and Chukchi Sea), a unique age peak
at ~90–110 Ma appears in nine surface samples (Figs. 2M-U), whereas only one sample
(LV83-37), close to the Laptev Sea, lacks this peak (Fig. 2L). These ~90-110 Ma zircons
are likely derived from the Okhotsk-Chukotka Volcanic Belt (~106–76 Ma) (Fig. 1)²⁹.
In the Bering Sea, the age peak at ~90-110 Ma also appears together with an additional
prominent peak at ~60 Ma (Fig. 2V), likely correlating with the magmatic belts within
the Yukon River basin that drains into the Bering Sea (Fig. 1)²⁵. On the North American
continent, surface sediments from the Mackenzie River Mouth lack zircon grains with
ages <1000 Ma, instead contain numerous zircons within the ranges of ~1000–1500 Ma
and ~1800–2000 Ma (Fig. 2W)²⁶, which are associated with the North American cratons
(Fig. 1).”

Additionally, we have created a multidimensional scaling (MDS) plot for the
surface and core samples in Fig. S7 (see below).

**Fig. S7. Multidimensional scaling (MDS) plots of zircon U-Pb ages from surface**
**and core samples of LV90-8-1 and LV90-9-1. (A) MDS plot of surface samples. The**
**numbers correspond to Figs. 1 and 2. (B) MDS plot of surface samples and core samples.**
**The numbers of core samples correspond to Fig. 3. MDS plots were generated in**
<https://isoplotr.es.ucl.ac.uk/>¹¹⁷.

However, the MDS analysis has both advantages and disadvantages in
distinguishing our surface samples. The advantage is that it quantifies the differences
in age distribution. The disadvantage is that the differences between the samples in the
MDS plot, reflected in the Kolmogorov-Smirnov (K-S) distance between samples, are
primarily influenced by the size or proportion of age peaks, rather than the presence of
a specific peak. We can see from Fig. S7 that, although samples with and without the
90-110 Ma peak are separated, the difference is not significant. To explain this
limitation, we calculated the K-S distance between surface samples 19 and 16, and
between 10 and 16 (See Fig. R1 below). The zircon age distributions for samples 19,

16, and 10 are shown in Fig. R1A, with the corresponding cumulative age distributions
 (CAD) in Fig. R1C. The K-S distance is defined as the largest distance between the
 CAD curves (Fig. R1C). Notably, both samples 16 and 19 exhibit a distinct 90-110 Ma
 peak (Fig. R1A) which is absent in sample 10. However, the K-S distance between
 samples 16 and 19 is large, even greater than that between sample 10 and 16 (Fig. R1D).
 This is due to the higher proportion of 90-110 Ma zircons in sample 19 (Fig. R1B and
 R1C).

 **Fig. R1. Using three surface samples to explain why the MDS plot fails to**
 **effectively differentiate the presence of a 90–110 Ma age peak. (A)** Zircon age
 distributions of surface samples 19, 16, and 10 (from Fig. 2). Distinct 90–110 Ma peaks
 are observed in samples 19 and 16, while absent in sample 10. **(B)** Proportion of the
 90–110 Ma peak (%) for these three samples, highlighting the significantly high 90–
 110 Ma age proportion in sample 19. **(C)** CAD plot, showing that the high proportion
 of 90–110 Ma ages in sample 19 causes its CAD curve to deviate from those of samples
 16 and 10. **(D)** MDS plot, demonstrating that the K-S distance between samples 19 and

16 is greater than that between samples 16 and 10.

Therefore, while the MDS plot provides valuable insights into the variability of
our samples, it is not an effective tool for distinguishing whether a sample contains the
90-110 Ma peak. As such, we have included the MDS plot in the supplementary
materials.

The identification of the age peak is primarily achieved through KDE curves with
a uniform bandwidth, as detailed in the Materials and methods section and in our
detailed response to Reviewer 2's first comment.

**Comment 5 on figure 3**

(1) The interglacials typically have a small amount of sand (IRD) but glacial periods
lack IRD until the very end. This pattern isn't mentioned in the text. What is the
explanation for this pattern? See other comments on the figure.

(2) Line 126: The boundaries based on a* seem arbitrary and in some places don't
adhere to your definition. This needs to be tightened up and reviewed to make it more
internally consistent.

(3) This fines upward (LV90-8-1, 25-40 cm), so why not a turbidite?

(4) Are the fine sediments the same composition throughout? If you are suggesting that
the sand provenance changes, does the fine fraction change as well? What insights can
this information provide?

**Reply:**

we updated Fig. 3 based on your suggestion above and added some explanations
accordingly.

 **Fig. 3. Core photo, color reflectance (a*), bulk density, sand content (> 63**
 **fraction) and detrital zircon sample positions of cores LV90-8-1 (A-C) and LV90-**
 **9-1 (D-F).** Glacial periods 1 to 6 with marine isotope stages (MIS) are shaded and
 indicated in green italics. The a* curve processed with a 5 cm moving average, where
 yellow points denote selected markers for glacial-interglacial boundary determination.
 The MIS is based on AMS ¹⁴C dating on total organic carbon and correlation to adjacent
 cores PS2757-8 and 29-GC1^{30,31} from the Lomonosov Ridge. Two conflicting age
 models are shown in the right box: one is based on stratigraphic correlation^{30,31} and
 optically stimulated luminescence (OSL) dating, and the other is using excesses in U-
 series isotopes^{32,33} (see materials and methods; Fig. S4). Sediments with high sand
 content (usually > 10%) were sampled for detrital zircon U-Pb dating, the positions of
 which are marked by hexagons with sample names.

**(1) Sand in interglacial and glacial periods:**

The sand content (>63 μm) during interglacial periods is typically < 5% (Fig. 3),
 and we hypothesize that these sands might be related to sea ice, because the threshold
 grain size for iceberg and sea-ice is 250 μm (see Reviewer 3's first comment below).

In addition, your comments regarding cold-based and warm-based ice sheets
provided valuable insight into our interpretation of why the sand-rich ESIS signal
appears at the end of the glacial period. We hypothesize that during the mid-glacial
stages, the East Siberian Ice Sheet's base remained below the pressure-melting point,
limiting basal sliding and erosive capacity, resulting in minimal IRD output (Kleman
and Glasser, 2007; Koppes et al., 2015). In contrast, during the late-glacial stages,
thawing at the ice sheet base increased mobility and erosive strength, leading to more
IRD output through calving icebergs (Rignot and Jacobs, 2002; Depoorter et al., 2013).
The appearance of IRD after GP 2 and late IGP 2 (both during MIS 3) suggests that the
collapse and melting of the ice sheet persisted into MIS 3, likely due to lower
temperatures compared to other interglacials like MIS 7, 5, and 1.

To ensure the logical coherence of the paper, we have added two brief
introductions on *Page 7, lines 137-141* “In cores LV90-8-1 and LV90-9-1, sand
contents remain low (<5%) during interglacial periods (IGP) but increases significantly
(>10%) towards the end or after each glacial period (GP) (Figs. 3C and 3F). An
exception occurs during the late IGP 2 (i.e., late MIS 3), which exhibits a high sand
content (~10%), possibly due to colder conditions during this period compared to MIS
5 and the Holocene in the Arctic Ocean^{4,7,9}.” And *Page 10, lines 214-217* “The presence
of a prominent ~90-110 Ma peak in the age distribution of coarse zircons (>150 μ m) in
sand-rich IRD layers suggests that they were transported by icebergs from East Siberia
(Fig. 4), likely during periods of ice melting in the late or post-glacial stages (Fig. 3).”

**(2) Glacial-interglacial boundary.**

Using a specific a^* value to determine the boundary between glacial and
interglacial periods is indeed arbitrary. Previous studies suggested that the maximum
and minimum values of a^* in Arctic sediments could represent peak glacial and
interglacial periods, respectively (e.g. Löwemark et al., 2014), but they did not provide
a convincing standard for defining the exact boundary. Therefore, we referred to the

approach used in LR04 for defining glacial-interglacial boundaries and set the midpoint
of rapid a^* changes as the boundary. We added this information in *Page 13, line 293-*
*298*, and marked the points in Fig. 3:

“The boundaries between glacial and interglacial periods in cores LV90-8-1 and
LV90-9-1 were determined based on abrupt changes in a^* values. Two points showing
the greatest variation in a^* values at the transition between glacial and interglacial
periods were identified (Fig. 3 and Supplementary Fig. S4), and the midpoint of the
interval between these points was defined as the boundary. Since no distinct abrupt a^*
change was observed between MIS 3 and MIS 2, the boundary was estimated using
AMS ^{14}C dating results.”

**(3) Turbidite fining-up feature**

The typical normal grading caused by turbidity currents in our samples has a
thickness ranging from 0.5 to 2 cm, and rarely exceeds 3 cm (see Fig. S5D and S5F
above). However, in the late GP 1 section, the variation in sand content reaches up to
15 cm (Fig. 3 and Fig. S5A), and there are no thin laminations or sharp basal features.
Therefore, we conclude that this is not a turbidite deposit.

**(4) Fine fraction provenance**

In fact, we have also conducted provenance analysis for the fine fraction using Sr-
Nd isotopes and clay minerals (ongoing work). Although these two methods do not
specifically distinguish East Siberian sources (see Supplementary Text S1), in layers
where the zircon age spectra indicate a greater contribution from East Siberia (at the
end of each glacial period, such as late GP 4), both Sr-Nd isotopes and clay mineral
analysis show a younger source region with the presence of illite, which is characteristic
of the East Siberian shelf clay minerals (see Fig. R2 below). Therefore, we can further
infer that the coarse and fine fractions of sediments in our cores share a similar

provenance and likely experienced the same transport mechanism, such as iceberg.
 However, these data are still being processed and are awaiting publication.

 **Fig. R2. Clay mineral content and Sr-Nd isotopic composition of < 2 µm fraction**
 **from LV90-8-1.** Note that, at end of each glacial period (grey shaded), the illite content
 is high, $^{87}\text{Sr}/^{86}\text{Sr}$ is low and ϵNd is high, which possibly indicate an East Siberian
 contribution. These data are being processed and more data need to be added.

**Comment 6 on figure 4**

(1) Visually and from the zircon proportion % on the right, B, D, G, and J look similar
 to each other, not just D and G. Additional statistical analyses are needed to better
 differentiate sample provenance and make the case for variable provenance.

(2) Line 178: Zircon data suggest not all sands have the same provenance. More clarity
 is needed about how you interpret provenance differences AND depositional process
 changes simultaneously.

**Reply:**

Although B (sample number 2-1 in the MDS plot above), D (3-1), G (5-1), and J
 (6-3) underwent different depositional processes (D and G are turbidites, while B and J
 are IRD), their zircon age spectra are quite similar (see Fig. S7 in the MDS plot above).
 This similarity is primarily due to their origin from source areas with similar zircon age

distributions. The turbidite samples are from the Laptev Sea and western East Siberian
Sea (see Fig. 2K and 2L, LV83-10 and LV83-37), while the IRD samples could have
received materials from the Kara Sea or the Laptev Sea, both of which were covered by
the Eurasian Ice Sheet. The Kara Sea, Laptev Sea and western East Siberian Sea are
indistinguishable because they share a 140-160 Ma peak (Fig. 2). Therefore,
establishing the relationship between depositional processes and provenance is
challenging in this study. Nevertheless, this does not affect the main conclusions of this
study.

**Other comments**

—**Line 58:** insert when.

**Reply:** we add ‘during the Quaternary glacial periods’ in *Page 3, line 58*.

—**Line 70-75:** Briefly state more about the +/- using DZ ages (e.g., Licht and Hemming,
2017). Depends on rock types, zircon fertility, etc., Why are there cases when a
multiproxy provenance approach may not be the best solution?

**Reply:** We have added the following sentence: “Zircon grains are highly resistant to
weathering, and their U-Pb age distributions in sediments are closely associated with
the magmatic history of their source regions²¹. Although zircon grains are less abundant
in carbonate and mafic rocks²², they are prevalent in felsic and intermediate igneous
rocks that are widely exposed across various Arctic source regions (Fig. 1). More
importantly, the age distribution of detrital zircons in sediments has the potential to
accurately distinguish between source regions with relatively small age differences (e.g.,
a few tens of millions of years), offering an advantage over traditional provenance tools,
such as mineral assemblages and geochemical isotopes (e.g., Sr-Nd isotopes), which
often produce mixed provenance signals^{10,11}.” in *Page 4, line 70-78*.

Therefore, this paragraph highlights two advantages of using DZ ages: first, their
resistance to weathering, and second, their ability to distinguish source regions with

small differences in bedrock ages. Despite potential fertility disadvantages, multiple
periods of felsic and intermediate igneous rocks around the Arctic make these regions
ideal for zircon provenance tracing (Fig. 1).

Furthermore, while a multiproxy provenance approach has been useful in many
studies, it may introduce bias in our case. Other provenance methods are difficult to
differentiate between the East Siberian and western-central Eurasian continents (see
Supplementary Text S1). The use of mineral assemblages and geochemical isotopes to
infer provenance can lead to mixed signals, increasing the complexity of source region
differentiation in our study.

—**Line 139:** Intense is too strong for many of the cores shown in this figure as
laminations are preserved.

**Reply:** We agree. We have rewritten this sentence by removing the term "intense" in
*Page 7, line 143-145:* “In contrast, the sand-rich layers during GP 1, 2, 4 and 6, as well
as late IGP 2, show a matrix-supported texture and gradational contacts with
bioturbation (Supplementary Fig. S5).”

—**Line 198:** The presence of the world's longest turbidite channel in the neighboring
basin suggests turbidites are not rare in this environment.

**Reply:** Yes, there are many turbidite channels around the Arctic Ocean basin, including
those in the northern Chukchi Sea and the northern East Siberian Sea (e.g., De Long
trough in Fig. S12 above). Many studies suggest that these turbidity currents around the
Arctic Ocean basin could be triggered by ice sheet advance (O'Regan et al., 2017). In
our cores, although at least two turbidites were identified (Fig. 3 and Fig. S5 above),
there is no 90-110 Ma age peak in the sediments of these turbidite layers (Fig. 4),
suggesting that their source is not East Siberia, but possibly from the surrounding
Laptev Sea. Thus, it is difficult to determine whether they are related to the ice sheet,
as many factors (such as ice sheets, earthquakes, etc.) can trigger turbidity currents.

—**Line 210:** Add reference(s)

**Reply:** we add two references in *Page 11, line 218* about the East Siberian Ice Sheet
evolution.

—**Line 231:** The albedo of sea ice and glacier ice would be similar, so albedo wouldn't
be largely different depending on the type of ice cover. This isn't a strong argument.

**Reply:** You are right. We have deleted this weak argument.

—**Line 237:** The paper would have higher impact if the concluding statements were
about the new contributions of this work (provenance) that can be used to advance
knowledge.

**Reply:** Thank you for your suggestion. In the conclusion, we have emphasized that our
provenance indicator can be applied to older sediments, which will help expand the
understanding of the development history of the East Siberian Ice Sheet. Please see
*Page 11, line 246-250:*

“Our new provenance indicator can be applied to older Arctic sediments,
facilitating a more comprehensive reconstruction of the East Siberian Ice Sheet^{19,56}.
Future research should aim to refine the extent of ice coverage in East Siberia and
reassess its implications for climate dynamics, as well as its potential influence on
human dispersal via the Bering Strait⁵⁹.”

—**Line 294:** I have not seen this approach used. If erosion shaped quartz grains as you
indicate with SEM, then to what degree are the edges of zircons also modified by
erosion. How might that affect your data?

**Reply:** The reason we focused on the edges of zircons is to avoid the influence of
xenocrystic cores, voids, and inclusions, which could lead to inaccurate zircon ages. I
understand your concern regarding edge erosion, but from the CL images (Fig. S13),

we observe that most zircons retain sharp edges, indicating their excellent resistance to
erosion. Even if some zircon experienced intense erosion that results in edge modified
and led to older zircon ages, this would only decrease the frequency of the 90-110 Ma
peak, making it less prominent, but it is unlikely to completely eliminate it, especially
given the large sample size in our analysis.

(A) LV77-16

(B) LV83-19

(C) LV83-37

(D) LV90-8-1 34-37 cm

(E) LV90-8-1 189-192 cm

(F) LV90-8-1 351-354 cm

Fig. S13. Example of cathodoluminescence (CL) images of zircon grains from samples from surface (A-C) and core LV90-8-1 (D-F). Red circles with a diameter of 37 μm represent the laser ablation points, which are chosen to avoid the xenocrystic core, voids and inclusions based on the CL images.

—**Figure S3.** What is the (n=X/Y) on the figure?

**Reply:** We have moved Fig. S3 to Fig. 2 in the new version. In all the updated figures,
we have added **legends** to clearly define what each symbol represents

—**Chronology** - I did not find a table detailing the radiocarbon dating results. This
needs to be published with the ages shown.

**Reply:** We have presented the radiocarbon dating results in Supplementary Table S3.

**Reviewer #2:**

**Overview**

This paper presents interesting results from detrital zircon U-Pb dating used to find
provenances of sediments in two marine cores from the Makarov Basin, central Arctic
Ocean. The linked provenances are in turn used to interpret the Arctic Ocean glacial
history; more specifically the authors propose specifically that zircon dates in samples
giving two peaks, ~90-100 Ma and ~150 Ma, are linked to an East Siberian source. Four
such intervals are found in the sediment cores, and those are linked to four distinct
glacial periods. The authors go on and suggest that this indicate that “the East Siberian
continental shelf was occupied by an ice sheet or glaciers during multiple glacial
periods.”

The existence of an East Siberian ice sheet has been raised by several authors, and
some evidences do exist, but the topic is far from resolved implying that it is very much
welcomed with new views. The paper is well written and figures are perhaps a bit
colorful, but clear. The dating methodology seems solid and the number of ages is
impressive.

However, I do have some overall concerns that to my mind should be addressed.

I think it is important to refer to previous results accurately, otherwise the scientific
history will be skewed. I kind of got stuck on the two main points raised below and did
not go much into details of the paper.

**Reply:**

We sincerely appreciate your welcome to our new view and new data. In the
revised version, we have thoroughly updated the manuscript, incorporating statistical
analyses to enhance the robustness of age peak identification and providing a detailed
review of previous studies. Nearly all figures have been updated, and the issue of overly
colorful visuals has been addressed to improve clarity and presentation. Please refer to
the reply to comments 1 to 3. In addition, we look forward to more comments in the
future if we could address your concerns at this stage.

**Comment 1 on identification of zircon age peak**

First, it is a bit difficult to evaluate how robust the provenance linkages are: How
significant are the zircon age peaks used as fingerprints of provenances?

This is hard to judge as Figure 2 has different scales on the y-axis, ranging from
0-20 to 0-80. For example, the peak for 149 Ma in F is represented by 6 dated grains,
while this would not show up as a peak in for example in B where there are more than
6 grains dated to this age as far as I can judge from the graph, which in B has a y-axis
scale of 0-80 instead of 0-20 as in F.

I would suggest here that a statistical cross-correlation between the records is used
to really show that there indeed is a robust correlation between the peaks in Figure 2.
Some of the peaks are obvious and clear, but some are not, simply as they are not
comprised of that many dated zircons. I would not have suggested this if all peaks stood
out clearly, and the fact that the peaks here are used to draw rather large conclusions on
the glacial history. So, if we could be convinced, the story would to my mind be much
stronger and one could leave this behind and go into details.

**Reply:**

To enhance the robustness of provenance identification using zircon age peaks, we
have made three updates to data processing in this study:

**(1) Utilizing statistical methods for identifying comparable age peaks:**

In the previous version, we visually determined the position of age peaks based on
the concentration of zircon ages. In the revised version, we employed a kernel density
estimation (KDE) method to quantitatively evaluate the concentration of ages, thereby
improving the robustness of source-to-sink comparisons. We added how this estimation
works in Materials and methods in *Page 16-17, line 344-371*:

“Zircon U-Pb age distribution are illustrated by kernel density plots (KDE) and
frequency histograms (Figs. 2 and 4). KDE is a non-parametric statistical method that
estimates the probability density function of zircon U-Pb ages⁷⁴, which quantitatively
identifies age peaks. It works by placing a kernel function, typically a Gaussian (normal)
function, at each measured zircon U-Pb age and then summing these kernels at each
position along the x-axis to create a smooth curve representing the density distribution.

The KDE at each position along the x-axis (x) is calculated using the following
formula⁷⁴:

$$f(x) = \frac{1}{nh} \sum_{i=1}^n K\left(\frac{x - x_i}{h}\right)$$

Where, $f(x)$ is the estimated density at each position along the x-axis, n is the
number of zircon U-Pb ages, h is the bandwidth that controls the width of the kernel,
K is the kernel function, x_i is the i -th zircon U-Pb age.

In practice, the kernel function K is typically chosen to be a Gaussian, defined as:

$$K\left(\frac{x - x_i}{h}\right) = \frac{1}{\sqrt{2\pi}} \exp\left(-\frac{1}{2}\left(\frac{x - x_i}{h}\right)^2\right)$$

Then the age peaks (local maximum) were identified in the curve if a given $f(x_j)$
satisfies the condition:

$$f(x_j) > f(x_{j-1}) \quad \text{and} \quad f(x_j) > f(x_{j+1})$$

Where x_{j-1} and x_{j+1} are the adjacent positions on either side of x_j along the
x-axis.

The bandwidth h is a critical parameter that determines the smoothness of the
estimated density function and the peak identification. The bandwidth determines the
range of influence that the kernel density function of each zircon age has on the x-axis.
A small value for h results in a more jagged, less smooth estimate, while a large value
for h leads to over-smoothing. Here, we apply a consistent bandwidth, 15 Myr, to both
surface and core samples. This allows for a direct comparison of the age peaks between
the core and the surface samples. The choice of 15 Myr as the bandwidth is based on
the minimal age difference of 30 Myr between the peaks at 90–110 Ma and 140–160
676 Ma. A bandwidth larger than 15 Myr would cause the kernel density functions of these
677 two age groups to overlap, making it difficult to distinguish between them. On the other
hand, a smaller bandwidth would make the curve less smooth and potentially distort the
peak shapes. Thus, 15 Myr is selected to balance smoothness and peak resolution
effectively.”

You can see that the bandwidth h is a critical parameter in KDE as it determines
the smoothness of the curve and the number of peaks. To ensure comparability across
samples, we use the same bandwidth (15 Myr) for all samples. We noted that, as long
as the bandwidth remains consistent, its size does not affect peak comparisons between
source and sink samples. For instance, a larger bandwidth might miss some peaks, but
the same bandwidth for both source and sediment samples ensures that these missing
peaks are not identified across all samples, avoiding any bias in source-to-sink
comparisons (see an example in Fig. R3 below).

**Fig. R3. Example demonstrating how the same bandwidth ensures source-to-sink**

**comparisons.** (A) Both surface sample LV77-31 and core sample late GP 4 exhibit a

prominent 90-110 Ma peak when the bandwidth $h=15$ Myr. (B) The 90-110 Ma peak is

not identified in either the source or sink samples when the bandwidth $h=30$ Myr, due

to the overlap of kernel functions from the 140-160 Ma and 90-110 Ma peaks. Thus,

the large h value, which is beyond the nearby age peaks, leads to over-smoothing.

**(2) Adequate zircon ages:**

The sample size of zircon ages also affects peak identification, with smaller sample

sizes likely to overlook minor peaks (Vermeesch, 2004; Pullen et al., 2014). In the

revised version, to ensure the >150 μm zircon age distribution (used to represent

iceberg-origin IRD, see Comment 1 from Reviewer 3) is robust, we added about 2500

additional zircon ages over the past three months. The explanation regarding the sample

size has been included in the Materia and method section in *Page 17, line 373-381*

“Additionally, the sample size of zircon ages influences peak identification, with

smaller sample sizes potentially overlooking minor peaks^{75,76}. In our samples, the 90–

110 Ma peak accounts for 2–8% of the age distribution (Table S2). Statistical analysis

and numerical simulations of zircon dating show that for sample sizes exceeding 500

grains, the likelihood of missing a 2% peak is negligible⁷⁷. Even with a smaller sample

size of 100 zircon grains, the 90–110 Ma peak is still clearly identifiable, with the

likelihood of missing a 2% peak around 13% and a 5% peak about 5%⁷⁷. In our case,
the zircon age distributions from >63 μm grains exceed 500 grains in sample size
(except for mid-GP6), while those from >150 μm grains are around 100 (Fig. 4).
Therefore, the zircon grains dated in our samples reliably capture the 90–110 Ma peak,
regardless of the sample size.”

**(3) Standardizing the y-axis scale:**

In the previous version, different scales on the y-axis created inconsistencies in
peak identification. While the KDE method already provides a quantitative approach to
peak identification, in the revised version, we unified the y-axis scales across all zircon
age spectra to facilitate visual peak identification (see Fig. 2 below): The scale for KDE
curves in left y-axis now ranges from 10^1 to 10^2 . Histograms were replaced with
frequency histograms with right y-scale ranging from 10^0 to 10^1 .

**(4) Displaying all samples instead of grouping by region:**

Following Reviewer 1's suggestion, we now display the zircon age spectra of all
individual samples in the updated Fig. 2, rather than grouping them by region. This has
two benefits: It clearly shows differences between individual samples; and, it provides
a more reasonable determination of age peaks, as small variations in peaks between
samples may be masked when grouped together.

Additionally, traditional methods such as multidimensional scaling (MDS) for
comparing zircon age spectra were not suitable for this study. This is because these
methods measure similarity based on the magnitude or proportion of age peaks, rather
than the presence or absence of peaks, which is the focus of this study. Please refer to
our response to Reviewer 1's comment 4 on Fig. 2 for more details.

Fig. 2. Detrital zircon U-Pb age distributions of surface samples from circum-

Arctic continental shelves. Sample locations are shown in Figure 1. ~90-110 Ma,

~140-160 Ma, ~220–360 Ma, ~420–360 Ma, and ~1750–2000 Ma zircon age peaks

are shaded. Each part of the circum-Arctic continental shelf has a distinct zircon age

distribution. The western and central Eurasian samples (A-K) lack the ~90-110 Ma

peak which is prominent in the East Siberian samples (**M-U**). The Bering Sea sample
(**V**) has a unique ~60 Ma peak. The North American sample (**W**) is characterized by
1000–1500 Ma zircons that are absent from all the Eurasian samples (**M-U**). Hence,
the ~90-110 Ma age peak exclusively fingerprint the East Siberian provenance.

**Comment 2 on ESIS evolution:**

Second, the authors over-simplify results from previously published studies in an
attempt to make their new results stand out as striking, contradictory and, perhaps,
provocative. This is not necessary, instead it would be better to embrace the earlier
studies and build on them accurately. They state, for example, that the “evolution of
Northern Hemisphere ice sheet was more complex than previously thought” . I’m not
sure I agree that the evolution is expressed to be particularly simple in the majority of
published papers; a potential ice sheet on the East Siberian Shelf has been raised in
several works, some of which the authors cite, but many more in addition. For example,
it can be found in earlier papers such as in Hughes et al. (1977, Nature), and papers by
Russian geographer Grosswald. Furthermore, they state that their results challenge the
notion of a “one-time glaciation in East Siberia occurred probably during MIS 6”
and refer to Jakobsson et al. (2016, Nature comm.). This paper is about a floating ice
shelf in the central Arctic Ocean, inferred from ice grounding at depths up to 1000 m.
It is correct that this paper points to glaciation of the East Siberian Shelf as an important
component for the ice shelf and should be cited as such, but it does not say that it only
ever occurred during MIS 6. It even ends with “Finally, we emphasize that our results
do not exclude the possibility that large ice shelves developed in the central Arctic
Ocean during other glaciations than MIS 6. On the contrary, it seems likely that ice
shelves were reoccurring components in most of the Quaternary glaciations considering
the land-locked nature of the Arctic Ocean.” In other words, it is the thickness and
extent of the ice shelf that seems to be larger in MIS 6 (assuming the age model in their
paper). Several more papers raise that ice shelves may have existed along the East

Siberian Shelf, see for example Ye et al. (2020,
<https://doi.org/10.1016/j.margeo.2020.106289>).

**Reply:**

Thank you for your feedback. In the revised version, we have presented a new
paragraph in the discussion section to reviewing previous studies and building on their
findings to highlight how our work advances the understanding of the East Siberian Ice
Sheet. Please see *Page 10-11, line 219-237*:

“However, the glaciation history of the East Siberia remains complex and highly
controversial for decades, largely due to uncertainties in chronology and fragmented
evidence. Three possible ice sheet configurations have been proposed. One suggests the
formation of an ice sheet on the East Siberian continent⁴⁹, with an ice shelf extending
over the East Siberian continent and into the shelf. The second scenario proposes an ice
sheet that covered the East Siberian continental shelf¹⁸, extending across the northern
East Siberian Sea, northern Chukchi Sea, and Chukchi Borderland (Fig. 1), with a
grounding line roughly at depths of ~300–1200 m (Supplementary Fig. S12). A third
possibility involves a massive ice shelf, approximately 1000 m thick, that covered the
entire Arctic Ocean, including the East Siberian continent³⁰. Some evidence, such as
the ages of the oldest sediments covering glacial deposits or landforms^{30,50-52} and
cosmogenic exposure ages of bedrock on shelf islands^{53,54}, suggests that the East
Siberian region may have been glaciated during MIS 6. Recent provenance studies also
support the existence of the East Siberian Ice Sheet^{15,55}. Here, by utilizing a distinct
IRD provenance indicator from East Siberia and accounting for chronology
uncertainties, our study demonstrates that East Siberian continent or continental shelf
underwent multiple glaciations during past glacial-interglacial cycles. According to the
available chronology, it is likely that the East Siberian experienced at least 2 to 3
glaciations during the last glacial-interglacial cycle (from MIS 5 to MIS 2, Figs. 3 and
4). As previous modelling studies have shown, once an ice sheet forms over the East
Siberian regions, it can trigger regional warming feedback that accelerate its melting,

leading to rapid fluctuations in ice sheet extent^{20,56}”

Based on this review, we outline two key advancements from our study:

**(1) Providing a relatively continuous developmental history of the East Siberian**
**Ice Sheet:**

Glacial geomorphic evidence confirms the existence of the East Siberian Ice Sheet
(ESIS) but does not provide precise constraints on its developmental timeline (e.g.,
Polyak et al., 2001; Niessen et al., 2013; Shen et al., 2024 and references therein). The
earliest sedimentary ages from glacial landforms (Jakobsson et al., 2016), along with
ages from glacial deposits on the New Siberian Islands and Wrangel Island (Gualtieri
et al., 2003; Stauch and Gualtieri, 2008; Nikolskiy et al., 2017; O'Regan et al., 2017),
offer some constraints but remain discontinuous. To establish a relatively continuous
record of the ice sheet's development, evidence must be sought from Arctic Ocean
sediments. Recent provenance studies of Arctic Ocean sediments have detected signals
potentially linked to the ESIS (e.g., Dong et al., 2020; Ye et al., 2020); however, the
provenance indicators used in these studies lack specificity for East Siberia and fine
particle do not directly signify iceberg transportation. In this study, we innovatively
identified provenance signals uniquely indicative of East Siberia and applied them to
Arctic Ocean sediment cores, thereby reconstructing a continuous history of the East
Siberian Ice Sheet's development.

**(2) Providing a robust tool for investigating the East Siberian Ice Sheet over longer**
**timescales:**

The specific provenance signals for East Siberia identified in this study can be
applied to older Arctic Ocean sediments in future research. This methodology holds the
potential to uncover the development history of the East Siberian Ice Sheet not only
throughout the Quaternary but also potentially extending into the Cenozoic. We added
this perspective in the conclusion paragraph in *Page 11, line 246-250*:

“Our new provenance indicator can be applied to older Arctic sediments,
facilitating a more comprehensive reconstruction of the East Siberian Ice Sheet^{19,56}.”

Future research should aim to refine the extent of ice coverage in East Siberia and
reassess its implications for climate dynamics, as well as its potential influence on
human dispersal via the Bering Strait⁵⁹.”

**Reviewer #3:**

**Comment 1 on differentiation of iceberg and sea ice origin:**

Han Feng et al. have amassed a large number of U-Pb zircon dates from several
surface samples from the East Siberian Arctic and sediment cores in the Arctic Ocean
and used these for provenance of ice-rafted detritus (IRD). They use the >63 micron
fraction to indicate iceberg IRD rather than sea ice IRD. Unfortunately, they did not
take note of more recent papers that show IRD found in modern sea ice samples contain
abundant sand greater than 63 microns (Darby, 2003; Darby et al. 2012, 2011; Reimnitz
et al. 1998). In fact, a large number of ice floes containing sand were deemed to be
anchor ice and these can contain even pebbles. They also use SEM texture on quartz
grains to indicate glacial iceberg IRD rather than sea ice IRD. However, IRD in Arctic
cores can be both sea ice and iceberg transported in the Pleistocene. There is no method
to determine that only icebergs transported IRD to the central Arctic in the past. So even
the presence of glacial features on quartz grains does not prove that only icebergs were
the mode of transport because core samples show subrounded features as well as
angular features. Furthermore, the general lack of nearshore and wave activity in the
Arctic is not conducive to well rounded grains. Thus the basic premise of the Han Feng
et al. paper that the core sediment sampled is iceberg derived is negated.

**Reply:**

We appreciate you raised important comments about the differentiation of iceberg-
or sea ice-origin IRD. We agree that “there is no method to determine that only icebergs
transported IRD to the central Arctic in the past”. However, in the revised version of
the manuscript, we focus on coarse grains (>250 µm for light minerals and >150 µm

for zircon) in IRD rather than >63 μm fraction. This approach is intended to better
highlight the characteristics associated with iceberg-origin IRD. Thus, we have refined
our argument to focus on zircons larger than 150 μm as evidence for iceberg transport.
This adjustment is based on the following reasoning:

**(1) Grain size differentiation between sea ice and iceberg transport:**

Sediments transported by sea ice are predominantly smaller than **250 μm** . While
sea ice can entrain grains larger than 63 μm via anchor ice processes, grains larger than
250 μm are very rare in modern sea-ice influenced sediments (e.g. Dethleff, 2005;
Eicken et al., 2005; Darby et al., 2011). This rarity is linked to the sea-ice entrainment
process, which typically occurs on the shelf area away from the beach and nearshore
zone (Reimnitz et al., 1998). Thus, like other previous studies (Reimnitz et al., 1998;
Darby et al., 2011), we agree that 250 μm can be regarded as a threshold to distinguish
sea ice from iceberg-transported sediments. We presented a new Fig. 5 (see below) to
show that surface sediments on the East Siberian continental shelf indeed contain very
rare >250 μm grains (except nearshore zone), while the core samples include 5-42%
grains >250 μm (also see a new Fig. S8). This implies that iceberg contributed >250
869 μm grains to the core site.

**Fig. 5. Grain size distribution of 100-500 μm grains from the surface samples on the Laptev Sea, East Siberian Sea and Chukchi Sea.** The
 vertical scale shows frequency per 10 μm bin normalized to total sand fraction (>63 μm). The gray histogram in the background of each sample
 represents the averaged grain size distribution from all sand-rich IRD samples in cores LV90-8-1 and LV90-9-1 (Fig. S8). Grains larger than 250
 874 μm account for 5% to 42% (~12% on average) in the sand fraction of IRD samples (Figs. S6 and S8), but less than 2% in almost all surface samples,
 except a near shore sample LV77-16. Grain size data of surface samples are from Ref. 62.

**Fig. S8. Grain size distribution of bulk and zircon grains from the sand-rich IRD**
 **samples in cores LV90-8-1 and LV90-9-1.** The vertical scale shows frequency per 10
 879 μm bin normalized to total sand fraction ($>63 \mu\text{m}$). The black curve represents the fitted
 average grain size distribution for all sand-rich IRD samples shown in Fig. 5 and Fig.
 6. See raw data in Source data S2 and S3.

**(2) Zircon grains larger than 150 μm are iceberg-origin:**

If sand grains larger than 250 μm are iceberg-origin rather than sea ice origin, then
 this size for zircon grains should be less. This is because of the zircon's high density
 ($\sim 4.7 \text{ g/cm}^3$), making it harder to be transported than light minerals (like quartz) under
 the same hydrodynamic conditions on the shelf (where sea ice entrains). Theoretical
 calculations on particle incipient motion and settling (provided in Supplementary Text
 S3) reveal that 250 μm quartz grains correspond to zircons of 113 and 158 μm ,
 respectively. Surface samples on the continental shelf seldom contain $>150 \mu\text{m}$ zircon
 grains (see a new Fig. 6 below). Therefore, we propose that sea ice is unlikely to
 entrain $>150 \mu\text{m}$ zircons from the shelf, and their presence in our core samples (Fig. 6
 and Fig. S8) strongly suggests iceberg transport.

**Fig. 6. Grain size distribution of 63-250 μm zircon grains from the surface samples on the Laptev Sea, East Siberian Sea and Chukchi Sea.**

The vertical scale shows frequency per 5 μm bin normalized to total sand fraction (>63 μm). The gray histograms in the background of each sample

represent the zircon grain size distribution from all sand-rich IRD samples in cores LV90-8-1 and LV90-9-1. Note that zircon grains larger than

150 μm account for ~6 % to ~18% (~13% on average) in the sand fraction of IRD samples (Fig. S8), but less than 3% in almost all surface samples,

except a near shore sample LV77-16.

Fig. 4. Detrital zircon U-Pb age distributions of > 63 μm and >150 μm fraction samples taken from sand-rich layers within cores LV90-8-1 and LV90-9-1. Sample positions are marked in Figure 3. Zircon U-Pb age peaks of ~90-110 Ma, ~140-160 Ma, ~220-360 Ma, 420-560 Ma and ~1750-2000 Ma are shaded in the zircon age distribution. The age peak of ~90-110 Ma occurs in the IRD samples of late GP 1 (A,

**K**), post GP 2 (**C, M**), late GP 4 (**E, F, O, P**) and late GP 6 (**H, I, R, S**), suggesting East
Siberian origin. ~90-110 Ma zircons are scarce in the IRD samples from late IGP 2
(**B, L**) and middle GP 6 (**J, T**), as well as in the two turbidite samples from late GP 3
(**D, N**) and late GP 5 (**G, Q**).

Over the past three months, we have conducted additional analyses, adding ~**2,500**
**new zircon U-Pb age** data to ensure statistical significance (≥ 100 grains per core
sample) for detrital U-Pb age distribution of zircons $>150 \mu\text{m}$. Our findings reveal that
the East Siberia-specific 90–110 Ma age peak is also present in this size fraction (see
Fig. 4 above), supporting the contribution of the East Siberian Ice Sheet during the
glacial periods.

This discussion about coarse sand and zircons in sand-rich IRD samples in cores
are the iceberg-origin have been rewritten in *Page 8-9, line 167-184*:

“Sea ice seems unlikely to be the main transporter of sand particles in the IRD
layers of cores LV90-8-1 and LV90-9-1 (Fig. 3), especially for coarser particles (>250
920 μm). The grains in these IRD layers are much coarser than those found in typical
modern sea ice sediments^{35,36}. Modern sea ice sediments seldom contain grains larger
than $250 \mu\text{m}$ ³⁶⁻³⁸, possibly because sea ice primarily incorporates sediments from shelf
areas away from the beach or nearshore zone (Fig. 5). In contrast, over 5% (~5-42%)
of the sand in the IRD layers exceeds $250 \mu\text{m}$ (Fig. 5, Supplementary Figs. S6 and S8).

Moreover, the much coarser zircon grains from the sand-rich IRD layers in cores
LV90-8-1 and LV90-9-1, compared to surface samples from the surrounding
continental shelves (Fig. 6), provide further evidence that the coarse sands in IRD layers
were not transported by sea ice. Zircon, being one of the densest minerals, is more
difficult to transport than lighter minerals like quartz or feldspar³⁹. In theory, sea ice is
unlikely to transport zircons larger than $150 \mu\text{m}$ (Supplementary Text S3), as shown by
the grain-size distribution of modern surface sediments (Fig. 6). When considering 150
932 μm as a threshold, ~6% to ~18% of zircons in the sand fraction of IRD layers exceed

this size (Supplementary Fig. S8), whereas almost all modern surface samples (except
those from near shore zone) rarely contain zircons larger than 150 μm (Fig. 6). The
presence of these coarser zircons in the sand-rich layers clearly indicates that a more
energetic transport agent beyond the sea ice, such as icebergs, is responsible.”

Furthermore, while we acknowledge that quartz micromorphology is a statistical
approach and cannot definitively confirm iceberg transport for sand-sized grains, we
use it as a supplementary line of evidence. Detailed descriptions of this method and its
results are included in the Supplementary Text S4.

**Comment 2 on source region:**

Secondly, they have no samples from the Axel Heiberg Island and Ellesmere Island
and Banks Island areas that were major outlets for North American icebergs from the
Laurentide and Innuitian Ice Sheets (Darby and Zimmerman, 2008). The only samples
from the Axel Heiberg and Ellesmere Island areas were from previous studies using Sr
- Nd provenance and these have a large overlap with other areas in the Arctic (see
Supplemental Materials in this ms).

**Reply:**

Thank you for pointing out the lack of direct sampling from Axel Heiberg Island,
Ellesmere Island, and Banks Island. While we acknowledge this limitation, we believe
our findings are still robust due to the following considerations:

**(1) Inference from Sr-Nd data:**

Although we do not have zircon U-Pb age data from the Arctic Archipelago, the
Sr-Nd isotope composition of sediments from this region (Fig. S1) indicates a
dominance of older rocks, with high Sr and low Nd isotopic values. These isotopic
signatures show no overlap with those of East Siberia, although there is some overlap
with Western Europe (see Fig. S1 below). Given the well correlation between Sr-Nd
isotopic signatures and zircon age distributions, it is reasonable to infer that sediments

from the Arctic Archipelago likely contain very few young zircons.

**Fig. S1. Sr-Nd isotopic values of surface sediments on the circum-Arctic**
 **continental shelves.** The $^{87}\text{Sr}/^{86}\text{Sr}$ (A) and ϵ_{Nd} (B) are effective provenance tools in
 distinguishing sources between North America and the entire Eurasia due to their
 distinctive values (C). The entire Eurasia (including East Siberia, central Eurasia, and
 western Eurasia) exhibits relatively higher ϵ_{Nd} (mostly -6 to -14) and lower $^{87}\text{Sr}/^{86}\text{Sr}$
 (mostly 0.71 to 0.72), whereas the North America show lower ϵ_{Nd} values (mostly < -12)
 and higher $^{87}\text{Sr}/^{86}\text{Sr}$ (mostly > 0.72). However, the ranges of Sr-Nd isotopic values of
 different shelves within the Eurasian shelf were overlapped (C). For example, both the
 East Siberian area (including the Chukchi Sea and the East Siberian Sea, which covered
 by proposed ESIS) and the Kara Sea (covered by EAIS) show high ϵ_{Nd} (mostly -6 to -
 9) and low $^{87}\text{Sr}/^{86}\text{Sr}$ (mostly < 0.712), thus Sr-Nd isotopic values cannot be regarded as
 an exclusive indicator of East Siberian provenance. Epsilon Nd values (ϵ_{Nd}) were
 calculated using chondritic values of $^{143}\text{Nd}/^{144}\text{Nd} = 0.512638$ ¹¹⁵. Data are from Ref. 10,
 11 and references therein. ESIS: East Siberian Ice Sheet¹⁸ (enclosed by the white
 dashed line); NAIS: North American Ice sheet⁶⁰; EAIS: Eurasian Ice Sheet⁶⁰; CS =
 Chukchi Sea; LS = Laptev Sea.

**(2) Constraints from iceberg transport pathways:**

For zircons from the Arctic Archipelago to reach the LV90-8 and LV90-9 core sites
via Inuitian Ice Sheet transport, they would have had to follow the Beaufort Gyre
circulation. Along this path, these icebergs would traverse regions influenced by the
Laurentide Ice Sheet, particularly its major outlets (See **Fig. R4** below). Given that the
Laurentide and Inuitian Ice Sheets generally developed synchronously on orbital
timescales (Darby and Zimmerman, 2008; Batchelor et al., 2019), it is highly likely that
sediments carried by Inuitian icebergs would be mixed with a substantial contribution
from Laurentide sources. This would introduce zircon age peaks typical of Laurentide
sources, particularly in the 1.0–1.5 Ga and 1.5–2.0 Ga ranges (Fig. 2W). However, our
zircon U-Pb age data show no significant 1.0–1.5 Ga peaks (Fig. 4), which strongly
suggests that sediments from the Arctic Archipelago, even if present, do not dominate
the zircon fraction in our studied cores.

**Fig. R4. Pathway of icebergs from Innuitian Ice Sheet to the site of LV90-8-1 and**
 **LV90-9-1.**

**Comment 3 on provenance indicator of minerals**

Finally, total IRD found in Fram Strait and cores near Denmark Strait (Darby and
 998 Zimmerman, 2008; Darby et al., 2017) show very little from the East Siberian Sea using
 a newer matching method for Fe Oxide fingerprinting that is overlooked by the authors
 of this submitted paper (Darby et al., 2015). This provenance Fe Oxide fingerprinting
 method is show to be accurate to less than 2% error in source determination based on
 nearly 40,000 grain analysis using micro probe analysis of Fe grains for 14 elements
 (Darby et al., 2015). Also it is true as pointed out by the authors that heavy mineralogy

is not the most effective provenance tool in the Arctic, but the mineralogy of hundreds
of circum-Arctic samples has indicated that there are differences between general areas
in the Arctic with regards to feldspar type and other light minerals (Myers and Darby,
2022). The authors need to revise their conclusions and make a better argument for
iceberg IRD, perhaps by showing that the zircon grains are much larger than 250
microns.

**Reply:**

We have carefully reviewed the newer matching method for Fe Oxide
fingerprinting. This provenance tool has been effective in tracing IRD in the Arctic,
whether in the Cenozoic (e.g., Darby, 2014; Tripathi and Darby, 2018) or Quaternary
sediments (e.g., Darby et al., 2012; Darby et al., 2017). However, as you mentioned,
studies in both the central Arctic and Fram Strait have shown minimal contributions
from the East Siberia (Darby and Zimmerman, 2008; Darby et al., 2017). For example,
less than 5 Fe grains (out of 100 grains) matched the East Siberian source in Darby
(2014) and Tripathi and Darby (2018), which suggest less than 2% (5/100) contribution
from the East Siberia.

Therefore, we acknowledged the effectiveness of other provenance indicators,
such as Fe-grains, in differentiating sediment sources across the Arctic Ocean in the
introduction, for instance in *Page 3, line 51-54*: "Provenance proxies, such as mineral
assemblages and isotope geochemistry, have been effective in differentiating sediment
sources between the continental shelves of North America and the entire Eurasia
(Bischof and Darby, 1997; Maccali et al., 2018; Myers and Darby, 2022)." At the same
time, we emphasized the difficulty of identifying East Siberian provenance in the
introduction, using statements in *Page 3, line 62-64*: "However, the absence of an
effective method to distinguish East Siberia sediments from those of other Arctic
continental shelves limits our understanding of the evolution of Northern Hemisphere
glaciation during past glacial-interglacial cycles. For example, whether and when an
Eastern Siberian ice sheet existed has been debated for over two decades." We have

also updated the relevant section in the Supplementary Text S1.

While our 90–110 Ma zircon peak is also relatively minor (e.g., ~2% in late GP1
which suggest ~2% contribution from the East Siberia, see Table S2), it remains
identifiable due to the large sample size of zircon U-Pb ages. We provide this
information in Materials and method in *Page 17, line 370-379*:

“Additionally, the sample size of zircon ages influences peak identification, with
smaller sample sizes potentially overlooking minor peaks^{75,76}. In our samples, the 90–
110 Ma peak accounts for 2–8% of the age distribution (Table S2). Statistical analysis
and numerical simulations of zircon dating show that for sample sizes exceeding 500
grains, the likelihood of missing a 2% peak is negligible⁷⁷. Even with a smaller sample
size of 100 zircon grains, the 90–110 Ma peak is still clearly identifiable, with the
likelihood of missing a 2% peak around 13% and a 5% peak about 5%⁷⁷. In our case,
the zircon age distributions from >63 µm grains exceed 500 grains in sample size
(except for mid-GP6), while those from >150 µm grains are around 100 (Fig. 4).
Therefore, the zircon grains dated in our samples reliably capture the 90–110 Ma peak,
regardless of the sample size.”

In addition, we appreciate your reference to the excellent work on light mineral
distributions in the Arctic (Myers and Darby, 2022). However, as with other minerals
and Sr-Nd isotopes, there remains no single mineral capable of uniquely identifying the
East Siberian provenance (See a new Fig. S3 below). For instance, while the East
Siberian Sea contains higher concentrations of plagioclase and tridymite, plagioclase is
also abundant in the Kara Sea, and tridymite concentrations are high in the Laptev Sea.
Similarly, East Siberia’s clay mineral composition shows elevated illite content, but this
cannot be reliably applied to IRD (mostly coarse sand) provenance tracing. These
clarifications are detailed in the Supplementary Text S1.

**Fig. S3. Quartz and feldspar distribution in Arctic Ocean surface sediments.**
 Redrawn from Ref. Myers et al., 2022. Note that the quartz and feldspar contents in
 East Siberia are similar to those in central Eurasia (Laptev Sea) and western Eurasia
 (Kara Sea). CS = Chukchi Sea; ES = East Siberian Sea; LS = Laptev Sea; KS = Kara
 Sea; BS = Barents Sea.

**References**

Batchelor, C. L., and J. A. Dowdeswell (2015), Ice-sheet grounding-zone wedges
 (GZWs) on high-latitude continental margins, *Marine Geology*, 363 65-92.

Batchelor, C. L., M. Margold, M. Krapp, D. K. Murton, A. S. Dalton, P. L. Gibbard, C.
 R. Stokes, J. B. Murton, and A. Manica (2019), The configuration of Northern
 Hemisphere ice sheets through the Quaternary, *Nature Communications*, 10(1),
 3713.

Culver, S. J., P. A. Bull, S. Campbell, R. A. Shakesby, and W. B. Whalley (1983),
 Environmental discrimination based on quartz grain surface textures: a statistical
 investigation, *Sedimentology*, 30(1), 129-136.

D. Dethleff, Entrainment and export of Laptev Sea ice sediments, Siberian Arctic.

*Journal of Geophysical Research: Oceans* **110**, C7009 (2005).

Darby, D. A. (2014), Ephemeral formation of perennial sea ice in the Arctic Ocean
during the middle Eocene, *Nature Geoscience*, 7(3), 210-213.

Darby, D. A., and P. Zimmerman (2008), Ice-rafted detritus events in the Arctic during
the last glacial interval, and the timing of the Innuitian and Laurentide ice sheet
calving events, *Polar Research*, 27(2), 114-127.

Darby, D. A., J. D. Ortiz, C. E. Grosch, and S. P. Lund (2012), 1,500-year cycle in the
Arctic Oscillation identified in Holocene Arctic sea-ice drift, *Nature Geoscience*,
5(12), 897-900.

Darby, D. A., J. T. Andrews, S. T. Belt, A. E. Jennings, and P. Cabedo-Sanz (2017),
Holocene Cyclic Records of Ice-Rafted Debris and Sea Ice Variations on the East
Greenland and Northwest Iceland Margins, *Arctic, Antarctic, and Alpine Research*,
49(4), 649-672.

Darby, D. A., W. B. Myers, M. Jakobsson, and I. Rigor (2011), Modern dirty sea ice
characteristics and sources: The role of anchor ice, *Journal of Geophysical*
*Research: Oceans*, 116(C9), C9008.

Depoorter, M. A., J. L. Bamber, J. A. Griggs, J. T. M. Lenaerts, S. R. M. Ligtenberg, M.
R. van den Broeke, and G. Moholdt (2013), Calving fluxes and basal melt rates of
Antarctic ice shelves, *Nature*, 502(7469), 89-92.

Dong, L., L. Polyak, Y. Liu, X. Shi, J. Zhang, and Y. Huang (2020), Isotopic fingerprints
of ice-rafted debris offer new constraints on Middle to Late Quaternary Arctic
circulation and glacial history, *Geochemistry, Geophysics, Geosystems*, 21(8),
e2020GC009019.

Eicken, H., R. Gradinger, A. Gaylord, A. Mahoney, I. Rigor, and H. Melling (2005),
Sediment transport by sea ice in the Chukchi and Beaufort Seas: Increasing
importance due to changing ice conditions? *Deep Sea Research Part II: Topical*
*Studies in Oceanography*, 52(24), 3281-3302.

Gualtieri, L., S. Vartanyan, J. Brigham-Grette, and P. M. Anderson (2003), Pleistocene

raised marine deposits on Wrangel Island, northeast Siberia and implications for
the presence of an East Siberian ice sheet, *Quaternary Research*, 59(3), 399-410.

Jakobsson, M., J. Nilsson, L. Anderson, J. Backman, G. Björk, T. M. Cronin, N.
Kirchner, A. Koshurnikov, L. Mayer, R. Noormets, M. O'Regan, C. Stranne, R.
Ananiev, N. Barrientos Macho, D. Cherniykh, H. Coxall, B. Eriksson, T. Flodén,
1108 L. Gemery, Ö. Gustafsson, K. Jerram, C. Johansson, A. Khortov, R. Mohammad,
and I. Semiletov (2016), Evidence for an ice shelf covering the central Arctic
Ocean during the penultimate glaciation, *Nature Communications*, 7(1), 10365.

Kleman, J., and N. F. Glasser (2007), The subglacial thermal organisation (STO) of ice
sheets, *Quaternary Science Reviews*, 26(5), 585-597.

Koppes, M., B. Hallet, E. Rignot, J. Mouginot, J. S. Wellner, and K. Boldt (2015),
Observed latitudinal variations in erosion as a function of glacier dynamics, *Nature*,
526(7571), 100-103.

Křížek, M., K. Krbcová, P. Mida, and M. Hanáček (2017), Micromorphological
changes as an indicator of the transition from glacial to glaciofluvial quartz grains:
Evidence from Svalbard, *Sedimentary Geology*, 358 35-43.

Löwemark, L., C. März, M. O'Regan, and R. Gyllencreutz (2014), Arctic Ocean Mn-
stratigraphy: genesis, synthesis and inter-basin correlation, *Quaternary Science*
*Reviews*, 92 97-111.

McCave, I. N., and I. R. Hall (2006), Size sorting in marine muds: Processes, pitfalls,
and prospects for paleoflow - speed proxies, *Geochemistry, Geophysics,*
*Geosystems*, 7(10), Q10N05.

McCave, I. N., and J. T. Andrews (2019), Distinguishing current effects in sediments
delivered to the ocean by ice. I. Principles, methods and examples, *Quaternary*
*Science Reviews*, 212 92-107.

Miller, M. C., I. N. McCave, and P. D. Komar (1977), Threshold of sediment motion
under unidirectional currents, *Sedimentology*, 24(4), 507-527.

Myers, W. B., and D. A. Darby (2022), A compilation of the silt and clay mineralogy

from coastal and shelf regions of the Arctic Ocean, *Marine Geology*, 454 106948.

Niessen, F., J. K. Hong, A. Hegewald, J. Matthiessen, R. Stein, H. Kim, S. Kim, L.

Jensen, W. Jokat, S. Nam, and S. Kang (2013), Repeated Pleistocene glaciation of

the East Siberian continental margin, *Nature Geoscience*, 6(10), 842-846.

Niessen, F., J. K. Hong, A. Hegewald, J. Matthiessen, R. Stein, H. Kim, S. Kim, L.

Jensen, W. Jokat, S. Nam, and S. Kang (2013), Repeated Pleistocene glaciation of

the East Siberian continental margin, *Nature Geoscience*, 6(10), 842-846.

Nikolskiy, P. A., A. E. Basilyan, and V. S. Zazhigin (2017), New data on the age of the

glaciation in the New Siberian Islands (Russian Eastern Arctic), *Doklady Earth*

*Sciences*, 475(1), 748-752.

O'Regan, M., J. Backman, N. Barrientos, T. M. Cronin, L. Gemery, N. Kirchner, L. A.

Mayer, J. Nilsson, R. Noormets, C. Pearce, I. Semiletov, C. Stranne, and M.

Jakobsson (2017), The De Long Trough: a newly discovered glacial trough on the

East Siberian continental margin, *Clim. Past*, 13(9), 1269-1284.

Polyak, L., M. H. Edwards, B. J. Coakley, and M. Jakobsson (2001), Ice shelves in the

Pleistocene Arctic Ocean inferred from glaciogenic deep-sea bedforms, *Nature*,

410(6827), 453-457.

Pullen, A., M. Ibáñez-Mejía, G. E. Gehrels, J. C. Ibáñez-Mejía, and M. Pecha (2014),

What happens when n= 1000? Creating large-n geochronological datasets with

LA-ICP-MS for geologic investigations, *Journal of Analytical Atomic*

*Spectrometry*, 29(6), 971-980.

Reimnitz, E., M. McCormick, J. Bischof, and D. A. Darby (1998), Comparing sea-ice

sediment load with Beaufort Sea shelf deposits; is entrainment selective? *Journal*

*of Sedimentary Research*, 68(5), 777-787.

Rignot, E., and S. S. Jacobs (2002), Rapid Bottom Melting Widespread near Antarctic

Ice Sheet Grounding Lines, *Science*, 296(5575), 2020-2023.

Shen, Z., C. Yang, T. Zhang, and Y. Xu (2024), A more complete and detailed glacial

history of the northwestern Chukchi margin—Implications for the existence and

evolution of the East Siberian-Chukchi ice sheet, *Quaternary Science Reviews*,
342 108915.

St. John, K. (2008), Cenozoic ice-rafting history of the central Arctic Ocean:
Terrigenous sands on the Lomonosov Ridge, *Paleoceanography*, 23(1), PA1S05.

Stauch, G., and L. Gualtieri (2008), Late Quaternary glaciations in northeastern Russia,
*Journal of Quaternary Science*, 23(6-7), 545-558.

Sugden, D. E. (1977), Reconstruction of the Morphology, Dynamics, and Thermal
Characteristics of the Laurentide Ice Sheet at its Maximum, *Arctic and Alpine*
*Research*, 9(1), 21-47.

Tripathi, A., and D. Darby (2018), Evidence for ephemeral middle Eocene to early
Oligocene Greenland glacial ice and pan-Arctic sea ice, *Nature Communications*,
9(1), 1038.

Vermeesch, P. (2004), How many grains are needed for a provenance study? *Earth and*
*Planetary Science Letters*, 224(3-4), 441-451.

Woodgate, R. A., K. Aagaard, R. D. Muench, J. Gunn, G. Björk, B. Rudels, A. T. Roach,
and U. Schauer (2001), The Arctic Ocean Boundary Current along the Eurasian
slope and the adjacent Lomonosov Ridge: Water mass properties, transports and
transformations from moored instruments, *Deep Sea Research Part I:*
*Oceanographic Research Papers*, 48(8), 1757-1792.

Ye, L., W. Zhang, R. Wang, X. Yu, and L. Jin (2020), Ice events along the East Siberian
continental margin during the last two glaciations: Evidence from clay minerals,
*Marine Geology*, 428 106289.

**Reply to the reviewers' comments on "Arctic zircon U-Pb ages reveal**
**multiphase glaciations in East Siberia during the late Quaternary" by**
**Feng et al.**

**Reviewer #1 (Remarks to the Author):**

**Overview:** The authors have made substantial revisions to this manuscript and
compared to the earlier submission, this paper makes a clearer case for coarser layers
of sediment from deep sea cores being of ice-rafted origin. I have some additional
comments that I'd like to see addressed. With those addressed, I think this work will
make a valuable contribution to the discussion of an East Siberian ice sheet.

**Reply:** We appreciate your positive assessment of our revised manuscript and your
constructive comments. Below we provide point-by-point responses to your comments,
which we have carefully addressed in the revised manuscript.

**Comment 1:** In Fig. 1, it is hard to see the distribution of rocks that are considered
sources to the offshore record. The magmatic rocks are highlighted, but they are not the
only ones that could be contributing zircons. There seems to be a wide distribution of
sedimentary rocks in this region through which the zircons may be recycled. State
whether there are DZ data from extensive Neogene sedimentary rocks or not, and if so
whether they contain grains 90-110 Ma. Also, the legend lists 'geological units', but time
periods are listed, not units. Rocks shown by time period are not particularly helpful
without additional information. Reconsider what kind of basemap may be more useful.
Zircon-bearing rocks along with the age population in them would be best, but creating

such a map would be a whole project by itself. This geographic region isn't my area of
expertise, so I'm not sure what may be available to use as an alternative. The lower left
1/3 of the map isn't really needed, so it could be cut off so the remainder could be
enlarged. There is a typo on the map = North Ameriaca should be fixed

**Reply:** Thanks for your suggestions. In Fig. 1, we chose to highlight only the magmatic
rocks because they represent the primary sources of detrital zircon in our study area.
Firstly, nearly all the zircons in our samples are of magmatic origin rather than
metamorphic, as indicated by their Th/U ratios—95% of the detrital zircons have Th/U >
0.1 (Rubatto, 2002), as shown in the newly added Fig. S14 below. Second, while it is
inevitable that shelf sediments include recycled zircons, the ultimate sources of these
recycled zircons are still the primary magmatic rocks in the source region. These
recycled zircons represent magmatic zircons that were temporarily stored in terrestrial
sedimentary rocks and were later re-eroded into the shelf area. Although they can
influence the age distribution of detrital zircons in shelf sediments, our study does not
rely on the overall shape of zircon age distributions to distinguish among circum-Arctic
source areas, but rather on the presence of the distinct 90–110 Ma age peak. For
example, regardless of the amount of recycled zircons present, sediments on the
western-central Eurasian continental shelf do not show a 90–110 Ma peak because
magmatic rocks of this age do not exist within those source areas (Fig. 1).

**Fig. S14. Th/U ratio of zircon grains in our surface (A) and core (B) samples**
**showing most of the zircon grains (~95%) are of magmatic origin (Th/U > 0.1)**
**(Rubatto, 2002).**

We add the interpretations in the first paragraph of **Supplementary Text S2**:

“The detrital zircon U-Pb ages in both surface and core samples analyzed in this study
show a close correlation with magmatic rocks in the respective source regions (Fig. 1).
Approximately 95% of the zircons in our samples have Th/U ratios greater than 0.1
(Fig. S14), indicating a predominantly magmatic origin (Rubatto, 2002). While
recycled zircons—magmatic zircons temporarily stored in continental sedimentary
rocks and subsequently re-eroded and transported to the shelf—can inevitably influence
the detrital zircon U-Pb age distribution, they do not obscure the magmatic signal
(Andersen et al., 2022). This is because the ultimate origin of recycled zircons remains
the magmatic rocks in the source area.”

Additionally, we appreciate your observation that “rocks shown by time period are not
particularly helpful.” Since the characteristic zircon ages of magmatic rocks are already
highlighted in Fig. 1, the geological ages of the rocks themselves are not directly
relevant to the focus of this study. Therefore, we have replaced the original background
with a DEM map, which also matches the bathymetric DEM used offshore. Please see
the new Fig. 1 below.

*Rubatto, D. Zircon trace element geochemistry: partitioning with garnet and the link*
*between U–Pb ages and metamorphism. Chemical Geology 184, 123-138 (2002).*

**Fig. 1. Regional setting of the circum-Arctic region.** The outer circle presents range
 of the circum-Arctic continental shelves, where the entire Eurasian continental shelf is
 divided into three parts: western, central, and eastern Eurasian (East Siberia) continental
 shelf. NAIS (North American Ice Sheet) and EAIS (Eurasian Ice Sheet) during LGM
 are enclosed by the white solid line⁷⁸; Proposed ESIS (East Siberian Ice Sheet) is
 enclosed by the white dashed line¹⁹. The times of intrusion or eruption of typical
 magmatic belts⁷⁹ are marked by black arrow with notes in colored rectangular²⁷. The
 colors of the surface sample points in this figure correspond to the fill colors of the
 zircon age distributions shown in Fig. 2. The red solid/dashed arrows indicate possible
 direction of Transpolar Drift⁶. Topographic and bathymetric data are from the GEBCO

Grid (<https://www.gebco.net/data-products/gridded-bathymetry-data>). LR:
Lomonosov Ridge; MB: Makarov Basin; CB: Chukchi Borderland; DLI: De Long
Islands; NSI: New Siberian Islands; WI: Wrangel Island; BS: Bering Strait; LGM: Last
Glacial Maximum.

Furthermore, although the lower left third of the map does not represent a source area
for our drill core samples, it is covered by the North American Ice Sheet. Given that we
refer to circum-Arctic ice sheets in the manuscript, and the North American Ice Sheet
is a major component of them, we prefer to retain North American (wrong spelling has
been fixed in Fig. 1) portion of the map.

**Comment 2:** Move Figure 2 to the supplement.

**Reply:** Identifying an exclusive provenance indicator for East Siberia is a key
component of this study and forms the basis for reconstructing the glacial history of the
East Siberian Ice Sheet using drill core data. We consider it useful to show readers that
the 90–110 Ma zircon age peak represents a distinctive East Siberian contribution.
Therefore, we prefer to keep Fig. 2 in the main text. Even so, we understand your
concern that the discussion of surface samples may be overly detailed. In response to
your comment—and in line with Reviewer 3’s suggestion—we have shortened the
section of the manuscript that discusses the surface samples. Please see *Page 5, Line*
*93-106*:

“The zircon age distributions in these surface samples show regional distinctions (Fig.
2). Notably, a distinctive zircon age peak at ~90-110 Ma clearly fingerprints the East
Siberian provenance (Figs. 2M-U) (zircon age peaks in other continental shelves are
discussed in Supplementary Text S2). This age peak is found in sediments from the East
Siberian continental shelf (including the East Siberian Sea and Chukchi Sea) but absent
in sediments from the western-central Eurasian (including the Barents Sea, Kara Sea
and Laptev Sea) and North American continental shelves, highlighting the difference

between these areas (Fig. 2). The ~90-110 Ma zircons are likely derived from the
Okhotsk-Chukotka Volcanic Belt (~106–76 Ma) that is exposed in East Siberia (Fig.
1)²⁷. Although sediments from the Bering Sea also contain a 90–110 Ma peak (Fig. 2V),
their unique ~60 Ma age peak—likely associated with magmatic belts in the Yukon
River basin (Fig. 1)—distinguishes them from those of the East Siberian continental
shelf (Figs. 2M-V). Therefore, an age distribution characterized by a peak at ~90–110
110 Ma, with the absence of a ~60 Ma peak, serves as an exclusive provenance indicator
for East Siberia, distinguishing it from western-central Eurasia, North America, and the
Bering Sea”

**Comment 3:** The chronological control on the timing of sediments is not strong. Only
one of the two cores was dated, and that chronology has numerous reversals. I think it
would be helpful to readers to move Fig. S4 (+/- the log for PS2757-8) in place of Fig.
3 in the main text. The points with the dates shown on cores 9-1 and 29-GC1 should be
in the main part of the paper.

**Reply:** This is an excellent suggestion. We have incorporated the content of Fig. S4
into Fig. 3 in the main text to more comprehensively show how the chronology of Arctic
sediments is established (please see the newly added Fig. 3 below). However, we fully
acknowledge that dating Arctic sediments remains a significant challenge. In a recent
and excellent study published in *Nature Communications*, Stein et al. applied multiple
methods to constrain the chronology of Quaternary Arctic sediments (Stein et al., 2025).
Even so, the dating of these sediments still heavily relies on stratigraphic correlation,
such as Mn stratigraphy, sediment color, density, and paleomagnetic inclination. The
few available absolute dates are also problematic—for example, most of the
radiocarbon ages are based on total organic carbon, which tends to produce older ages
and reversals (Suzuki et al., 2021); OSL dating is hindered by poor bleaching conditions

due to sea ice cover (West et al., 2021); and Th-Pa isotope dating still suffers from
considerable uncertainties (Purcell et al., 2022).

Therefore, to account for chronological uncertainties, we present two alternative age
models in the manuscript. This approach is similar to that of Stein et al. (2025), who
proposed both an old and a revised age model for core PS2758-8 in their Supporting
Information (please see their Fig. S4 below). As Reviewer 3 aptly noted, “the
stratigraphy of the two cores is about as good as it will ever get given the data at hand.”
Despite these limitations, we emphasize that the chronological uncertainty does not
significantly affect our main conclusion regarding the multiphase glaciation history of
the East Siberian Ice Sheet. We anticipate that future improvements in Arctic sediment
dating techniques will allow for a more precise reconstruction of this glacial history.

[REDACTED]

**Figure S4 from Stein et al. (2025).** For core PS2758-8, two age models have been

proposed in their paper, represented by the blue and red MIS labels, respectively. The
dark grey layer at the bottom could correspond to MIS 6, MIS 8, or MIS 10, but its
precise age remains uncertain.

*Stein, R. et al. A 430 kyr record of ice-sheet dynamics and organic-carbon burial in the*
*central Eurasian Arctic Ocean. Nature Communications* **16**, 3822 (2025).

*Suzuki, K., Yamamoto, M., Rosenheim, B. E., Omori, T. & Polyak, L. New radiocarbon*
*estimation method for carbonate-poor sediments: A case study of ramped pyrolysis*
*¹⁴C dating of postglacial deposits from the Alaskan margin, Arctic Ocean.*
*Quaternary Geochronology* **66**, 101215 (2021).

*West, G., Alexanderson, H., Jakobsson, M. & O'Regan, M. Optically stimulated*
*luminescence dating supports pre-Eemian age for glacial ice on the Lomonosov*
*Ridge off the East Siberian continental shelf. Quaternary Science Reviews* **267**,
*107082 (2021).*

*Purcell, K., Hillaire-Marcel, C., de Vernal, A., Ghaleb, B. & Stein, R. Potential and*
*limitation of ²³⁰Th-excess as a chronostratigraphic tool for late Quaternary*
*Arctic Ocean sediment studies: An example from the Southern Lomonosov Ridge.*
*Marine Geology* **448**, 106802 (2022).

Fig. 3. Core photo, color reflectance (a^*), bulk density, sand content ($> 63 \mu\text{m}$) and detrital zircon sample positions from cores LV90-8-1 (A-C) and LV90-9-1 (D-F), with stratigraphic correlations to 29-GC1 (G) and PS2757-8 (H, I). The thumbnail show locations of these cores. The a^* curves²⁹ of LV90-8-1 and LV90-9-1 were processed with a 5 cm moving average, where yellow points

denote selected markers for glacial-interglacial boundary determination. Glacial periods 1 to 6 are shaded and indicated in green italics.
The chronology of core LV90-8-1 and LV90-9-1 is based on AMS ^{14}C dating (green dots) on total organic carbon²⁹ and correlation to
adjacent cores PS2757-8 and 29-GC1^{30,31} from the Lomonosov Ridge. Bulk density stratigraphic tie points α_1 to α_5 from core PS2757-8
and 29-GC1^{30,31} are identified in core LV90-8-1 and LV90-9-1²⁹, indicating good correlation between these cores. Red dots with age text
in core 29-GC1 are optically stimulated luminescence (OSL) dating results⁶⁵. The occurrence of dinocyst *Operculodinium centrocarpum*
from PS2757-8 is marked by green bar that indicate MIS 3 and MIS 1⁶⁶. The variations in $^{230}\text{Th}_{\text{xs}}$ for core PS2757-8 are shown with its
“ $^{230}\text{Th}_{\text{xs}}$ extinction age” (~231 ka, early MIS 7) at depth of $\sim 5.9 \pm 1.5$ m^{32,33}. The blue italic “MIS” refers to the age model from Ref. 30
and 31, while the orange italic “MIS” refers to the age model from Ref. 32 and 33 (see Materials and methods). Sediments with high sand
content (usually > 10%) were sampled for detrital zircon U-Pb dating, the positions of which are marked by hexagons with sample names.
NB: Nansen Basin; AB: Amundsen Basin; LR: Lomonosov Ridge; MR: Mendeleev Ridge; CB: Canada Basin; MB: Makarov Basin; AP:
Alris Plateau; CP: Chukchi Plateau; NWR: Northwind Ridge.

**Comment 4:** Fig S5. It is unusual that with all the granules/pebbles shown in A and E
 (in particular) there is nothing in the core photos indicating their presence. If there are
 x-radiograph images showing the core with the larger size fractions in situ, they should
 be included.

**Reply:** We have included X-radiograph images in the revised Fig. S4 to address this
 comment. Although the resolution of our X-radiographs is relatively low (0.5 mm) and
 the scanned width is limited (~3 cm), they are still sufficient to capture the presence of
 IRD clasts with diameters of 3–5 mm. We have annotated these features in the figure
 accordingly.

**Fig. S4. Core photo, X-ray radiograph and sand content of typical sand-rich layers**
 **in core LV90-8-1 (A-F) and LV90-9-1 (G, H).** Sample positions of detrital zircon U-
 Pb dating are marked by hexagon with sample name below. Images of granules and
 pebbles found in the samples with positions are shown below. The white error bar

represents 2 mm. (A, B, C, E) Homogeneous composition with bioturbation (mottling);
The gradational contacts between adjacent beds indicate continuous deposition of
sediments, in which sands are typical ice rafted debris (IRD); (D, F) Thin lamination
with normal-graded sequence and sharp base indicating turbidite deposition. (G) Voids
left by fallen granules and pebbles indicate IRD deposition. (H) Homogeneous
composition with no bioturbation or beddings. GP = Glacial period. IGP = Interglacial
period.

**Comment 5:** The data in the right column of Figure 4 would be better in the supplement
and replaced with Fig. S7. Based on results in Fig. S7, I recommend the authors add
more nuance to their interpretations of the implications of this work. Samples 4-2 and
4-1 are convincingly like East Siberia sources, samples 2-1 and 5-1 are not, and the
remainder seem to indicate varying relative inputs from East Siberia and Central
Eurasia. Three possible ice sheet configurations, which configurations could explain
the variations in input/relative strength of an East Siberian source?

**Reply:** We have moved the right-hand panel of Fig. 4 to the supplementary materials
and replaced it with the MDS plot (originally Fig. S7), as suggested. Please see the
newly added Fig. 4 below. Thank you for your recommendation to add interpretation
regarding the relative input from East Siberia and western-central Eurasia. In fact, in an
earlier version of the manuscript, we did interpret the observed provenance changes in
terms of different configurations of the East Siberian Ice Sheet—consistent with the
idea of three possible ice sheet scenarios, as you proposed.

**Fig. 4. Detrital zircon U–Pb age distributions of sand-rich layers from cores LV90-**
 **8-1 and LV90-9-1, with multidimensional scaling (MDS) plots comparing core and**
 **surface samples.** Sample positions are marked in Fig. 3. Zircon U-Pb age peaks of ~90-
 110 Ma, ~140-160 Ma, ~220–360 Ma, 420–560 Ma and ~1750–2000 Ma are shaded in
 the zircon age distribution. The age peak of ~90-110 Ma occurs in the IRD samples of
 late GP 1 (A), post GP 2 (C), late GP 4 (E, F) and late GP 6 (H, I), suggesting East
 Siberian origin. ~90-110 Ma zircons are scarce in the IRD samples from U-late IGP 2
 (B) and middle GP 6 (J), as well as in the two turbidite samples from late GP 3 (D)
 and late GP 5 (G). Sample numbers of surface and core samples in the MDS plots
 correspond to those shown in Figs. 1–3. MDS plots were generated in

<https://isoplotr.es.ucl.ac.uk/>.

However, we eventually decided not to take this approach, as such interpretations are
not entirely robust. This is because variations in Arctic Ocean circulation can
significantly affect the detrital zircon signal from East Siberia in sediment cores. For
instance, even if the East Siberian Ice Sheet covered a large area, a change in the
direction of the Transpolar Drift (indicated by the dashed arrow in Fig. 1) could still
result in a diminished East Siberian contribution in the sediments.

We therefore chose to take a more conservative interpretation and limited our
conclusion to the presence (rather than the extent) of the East Siberian Ice Sheet. We
are currently exploring how Arctic circulation influences sediment provenance using
additional methods, such as Sr-Nd isotopic tracers of clay mineral, as mentioned in our
1st round of response.

**Comment 6:** Clarify language around ice shelves. Icebergs need not be sourced from
an ice shelf, would a tidewater margin have been possible in this region? We know from
Antarctica that large ice shelves usually yield ice bergs with minimal entrained
sediment as most melts out near the grounding line. Is there geomorphological evidence
for grounding zone wedges on the East Siberian shelf that should be brought into this
discussion?

**Reply:** Thank you for pointing this out. We have corrected our language regarding
iceberg origins to state that “icebergs form during calving from an ice shelf or ice cliff.”
In *Page 10, Line 209-210*.

The extent of the grounding line of the East Siberian Ice Sheet is shown in
Supplementary Fig. S12 and described in the main text (*Page 10, Line 216-218*): “The
second scenario proposes an ice sheet that covered the East Siberian continental shelf,
extending across the northern East Siberian Sea, northern Chukchi Sea, and Chukchi

Borderland (Fig. 1), with a grounding line roughly at depths of ~300–1200 m
(Supplementary Fig. S12).”

Given that our core sites are located in water depths greater than 2000 m, they fall
outside the expected grounding zone wedge area (Fig. S12). Therefore, the IRD found
in the cores is most likely derived from icebergs rather than from subglacial deposition
near the grounding line.

**Reviewer #2 (Remarks to the Author):**

**Overview:**

Dear authors, I find that that you have done a thorough revision of the manuscript. I
have only a few minor comments to be considered below.

**Reply:** Thank you for your positive comments. We appreciate your recognition of our
efforts in revising the manuscript and will carefully address the remaining comments.

**Line 57:** Mixture of terms. Eurasian Basin includes Amundsen and Nansen basins,
while Makarov Basin is a subset of Amerasian Basin. I suggest stating Amundsen Basin
and Nansen Basin instead of Eurasian Basin, to be consistent with the nomenclature use
for the Amerasian side. I propose also changing the map to reflect this.

**Reply:** Thank you for this careful and helpful suggestion. To ensure consistency in
terminology, we have revised the sentence on *Page 3, Line 63-66* to:

“Previous provenance studies of IRD have identified iceberg surges originating from
ice sheets on Eurasian or North American continental shelves in the central and eastern
Arctic Ocean (including the Lomonosov Ridge, Makarov Basin, Amundsen Basin, and
Nansen Basin; Fig. 1) during the Quaternary glacial periods.”

We have also updated the map in Fig. 1 accordingly to reflect this consistent
nomenclature.

**Line 65:** Note sure if reference 20 should be used, it is a paper that did not reach full
publication, and remained in discussion. Why not include some older reference where
it really began to be discussed:

Hughes, T.J., Denton, G.H., Grosswald, M.G., 1977. Was there a late-Würm Arctic ice
sheet? Nature 266, 596–602. <https://doi.org/10.1038/266596a0>

**Reply:** Thank you for providing this reference, we have replaced the reference 20 by
Hughes et al. (1977) here.

**Line 122:** Consider writing “...interpreted to represent an oxidizing environment with
reduced sea ice during interglacial periods.” I would not go so far as to say it signifies
seasonal ice only. There is no reference to back this up? The Mn cycles we first used as
proxy of interglacial or interstadial conditions in: Jakobsson, M., Lovlie, R., Al-Hanbali,
H., Arnold, E., Backman, J., Mörth, M., 2000. Manganese and color cycles in Arctic
Ocean sediments constrain Pleistocene chronology. *Geology* 28, 23–26.
[https://doi.org/10.1130/0091-7613\(2000\)28<23:MACCIA>2.0.CO;2](https://doi.org/10.1130/0091-7613(2000)28<23:MACCIA>2.0.CO;2)

**Reply:** We have revised this sentence to “Similar to most sediment cores from the
Arctic Ocean, the brownish/reddish (higher a*, redness) layers in cores LV90-8-1 and
LV90-9-1 are enriched in manganese (Mn) from circum-Arctic rivers, interpreted to
represent an oxidizing environment with reduced sea ice during interglacial periods...”
in *Page 5-6, Line 110-113*, and also add your suggested reference (Jakobsson et al.,
2000).

**Line 131:** “...we propose”... sounds like the two age models are proposed. Consider
writing “we put forward two alternative..”.....

**Line 186:** Should be “sea-ice origin”, not “sea ice-origin”

**Line 2013.** “Icebergs form during calving from an ice shelf....” An ice margin
extending into the ocean does not always have a floating extension in the form of an ice
shelf, it may also end in an ice cliff. So to be more accurate I suggest writing “Icebergs
form during calving from an ice shelf or ice cliff - that is....”

**Reply:** Thank you. We have revised these expressions in the revised manuscript.

**Line 201:** I do not understand the use of the references here: “This implies that the East

Siberia region was once glaciated, making an ice-free scenario
unlikely^{18,30}”.

If the references are moved to after ...glaciated, I believe it would be correct
considering the content of the papers referred to: “This implies that the East Siberia
region was once glaciated^{18,30}, making an ice-free scenario unlikely”.

**Reply:** We have moved the reference in the middle of the sentence as you suggest. This
change would be better.

**Line 225:** I suggest a slight modification to make it accurate with respect to what was
proposed in the paper referred to: “A third possibility involves a massive ice shelf,
approximately 1000 m thick, that covered the entire Arctic Ocean, and which may have
been a floating extension of an East Siberian ice sheet³⁰”

**Reply:** We have revised this sentence as you suggest.

**Reviewer #3 (Remarks to the Author):**

**Overview:** This submission is well written, but it could be tightened up and shortened
to make it more readable. It deals with an important issue, i.e., did a large ice sheet exist
on the E Siberian Shelf during low sea level cold intervals as far back as MIS 6 if the
stratigraphy is correct. The paper presents a large number of zircon dates that are
valuable for future Arctic provenance studies.

**Reply:** Thank you for your constructive comments and positive assessment. We
appreciate your suggestion to tighten and shorten the manuscript, such as *Page 5, line*
*93-106*, to improve readability. In the revised version, we have carefully edited the text
to make it more concise and clearer, while ensuring that key interpretations and data
remain well supported.

**Comment 1:** There are very important assumptions made that are not stated and need
to be addressed. Primarily, the question of whether the IRD and zircons found in the
two cores studied are glacial or sea ice-rafted. Studies that show the importance of
anchor ice are overlooked or mistaken for suspension freezing entrainment. The
contention that any sea ice-rafted sediment to the core sites is entrained by suspension
freezing is just not true. Previous studies show that extensive areas of sediment-laden
sea ice in the central Arctic are anchor ice. These ice floes contain 100 to a thousand
340 times more sediment than sea ice with sediment entrained by suspension freezing. Both
the HOTRAX and LOMROG expeditions in 2005 and 2007 found 100's of kilometers
of sea ice with high concentrations of sediment including large benthic shells and
sediment coarser than 250 microns. Thus by down-playing sea ice-rafting, the paper is
heavily biased toward iceberg transport.

**Reply:** Thank you for your helpful comment and for highlighting the importance of

anchor ice in sediment transport. We fully agree that anchor ice can entrain substantial
amounts of sand-sized particles ($>63\ \mu\text{m}$), and therefore cannot be ruled out as a
potential contributor to IRD in our cores. In response, we have revised the manuscript
to adopt more neutral and precise language regarding sediment transport mechanisms,
avoiding any unintended bias toward iceberg transport. For example, in the abstract, we
now write: “Central Arctic Ocean sediments from at least four glacial periods contain
coarse zircon grains that bear a diagnostic $\sim 90\text{--}110\ \text{Ma}$ age peak, **most likely** indicating
iceberg transport of debris from East Siberia”

**Importantly**, the possibility of sea ice rafting **does not** affect our key conclusion. As
discussed in our first-round revision, both theoretical and empirical evidence support
that grains $>250\ \mu\text{m}$ (corresponding to $>150\ \mu\text{m}$ zircons) are predominantly transported
by icebergs. These coarse zircon grains carry a clear East Siberian provenance signal
(Fig. S11) and supports a glacial (i.e., iceberg) origin linked to the East Siberian Ice
Sheet.

We would also like to note that the modern sea-ice data from the HOTRAX and
LOMROG expeditions that you cited actually support our interpretation. While these
studies reported widespread sediment-laden sea ice—much of it likely formed via
anchor ice—the actual grain-size data show that particles $>250\ \mu\text{m}$ are extremely rare.
In Darby et al. (2011), which analyzed 39 modern sea-ice samples, the mean grain size
was $\sim 17\ \mu\text{m}$, and the average of the 95th percentile grain sizes was $\sim 53\ \mu\text{m}$ (please see
their Fig. 4, reproduced below). Only three samples had 95th percentile sizes exceeding
$200\ \mu\text{m}$, and none exceeded $250\ \mu\text{m}$.

These observations are consistent with grain-size data from surface sediments on the
East Siberian shelf (our Fig. 5), which also show a scarcity of particles $>250\ \mu\text{m}$.
Therefore, although anchor ice can entrain seabed sand, it appears unlikely to be the
main transporter of grains $>250\ \mu\text{m}$ in the central Arctic Ocean.

.

**Figure 4 from Darby et al. (2011).** Texture of bulk dirty ice samples. The cluster of
most samples with sediment less than about 100 μm including L4 containing the shells.

Based on the above analysis, we have rewritten the first paragraph of the Discussion
section (*Page 7–8, Lines 154–165*). The newly added objective phrasing is indicated in
bold below:

“Arctic IRD can be transported by **either** icebergs or sea ice^{8,35}. However, sea ice **seems**
**unlikely** to be the main transporter of sand particles in the IRD layers of cores LV90-
8-1 and LV90-9-1 (Fig. 3), **especially** for coarser particles ($>250 \mu\text{m}$). The grains in
these IRD layers are much coarser than those found in typical modern sea ice
sediments^{36,37}. Although sea ice can incorporate seabed particles $>63 \mu\text{m}$ through the
anchor ice mechanism^{38,39}, previous studies have shown that modern sea-ice sediments
seldom contain grains larger than $250 \mu\text{m}$ ^{37,39,40}. This is likely because sea ice primarily
incorporates sediments from shelf areas away from the beach or nearshore zone, where
the proportion of $>250 \mu\text{m}$ grains is less than 2% (Fig. 5). Even during glacial periods
with lower sea levels, the coarse fraction on the shelves remained as low as it is today

(Supplementary Fig. S6). In contrast, over 5% (~5–42%) of the sand in the IRD layers
exceeds 250 μm (Fig. 5, Supplementary Figs. S5 and S7), suggesting a different
transport mechanism—**most likely icebergs.**”

This paragraph reflects a more nuanced interpretation: while sea ice may explain some
portion of the IRD (particularly in the 63–250 μm range), grains larger than 250 μm are
better explained by iceberg rafting.

*Darby, D. A., Myers, W. B., Jakobsson, M. & Rigor, I. Modern dirty sea ice*
*characteristics and sources: The role of anchor ice. Journal of Geophysical*
*Research: Oceans* **116**, C9008 (2011).

**Comment 2:** Feng et al. point out that >250 micron quartz has some glacial features
and that the texture of sediments on the Modern E Siberian Shelf are too fine-grained
to supply zircons >150 or 250 microns. This is all true but it is based on the assumption
of whether this shelf was covered by the same sediment that was fine-grained in texture
during low sea level stands as it is today. This assumption needs to be pointed out and
addressed. Surely, as sea level dropped and river gradients increased, coarser sediment
would be transported to the shelf. This could extend to the glacial shoreline where this
coarser sediment could be entrained by anchor ice.

**Reply:** We appreciate your recognition of our efforts in analyzing the >250 μm grain-
size fraction. Regarding your concern about whether sediment on the East Siberian shelf
was coarser during glacial lowstands compared to today, we compiled grain-size data
from five sediment cores collected from the Laptev and East Siberian Seas (please see
the newly added Fig. S6 below), spanning the last deglaciation and Holocene. These
records show no significant correlation between sea-level changes and grain-size
variations: coarse-grained fractions remain consistently low (~2–4 %) during both low
and high sea-level periods, although occasional peaks do occur—likely related to

increased river discharge during warmer Holocene intervals.

**Fig. S6.** Sand content (>63 μm, >150 μm and >250 μm) in cores from the
 continental shelves of the East Siberian Sea and Laptev Sea, showing no clear
 relationship with sea level rise since ~17 ka. Four time slices (~17 ka, 10 ka, 8 ka,
 and 2–0 ka) are shaded, with corresponding sea levels and core depths indicated in
 parentheses. The inset shows the core locations and topo-bathymetric maps at each time
 slice. The sand content of core PS2458¹²⁰ remained stable during rapid sea-level rise
 (~17–9 ka) with no significant decrease during the highstand period (~6–0 ka). The
 sand content of core LV83-16¹²¹ stayed consistently low since 12 ka, except for a >150
 425 μm peak at ~5.5 ka. The core LV83-32¹²¹ exhibited a distinct sand content peak around
 426 4 ka. The sand content of core LV77-36¹²² showed a gradual increase since ~8 ka. The
 427 core LV77-41¹²² maintained low sand content levels since ~7 ka.

Based on these observations, we infer that even during glacial periods with substantially
 lower sea levels, the East Siberian shelf likely did not supply large amounts of coarse

material (e.g., >250 μm) to anchor ice, similar to present-day conditions. While in
theory a lowered sea level could increase river gradients and enhance the transport of
coarser sediments, the East Siberian shelf is one of the widest and flattest continental
shelves in the world. Its geomorphological characteristics suggest inherently low
efficiency in transporting coarse sediments across long distances. Accordingly, we
added the following statement in the revised manuscript (*Page 8, Line 161-163*): “Even
during glacial periods with lower sea levels, the coarse fraction on the shelves remained
as low as it is today (Fig. S6).”

**Comment 3:** Another problem with the SEM quartz microfeatures of glacial origin is
that there is no way to determine that the quartz grains with glacial features were
derived from the E Siberian Shelf and not from sources like North America that were
glaciated. Other studies clearly show that there are multiple sources in sediment cores
in the central Arctic.

**Reply:** We agree that SEM analysis of quartz microfeatures cannot be used to determine
the provenance of the grains. However, our primary purpose for using SEM was to
assess the relative contribution of glacial versus sea ice transport mechanisms.
Provenance determination in our study relies on detrital zircon U-Pb geochronology,
which provides higher resolution for distinguishing sediment sources.

**Comment 4:** Because anchor ice entrains any sediment size on the seafloor, the
extensive discussion of incipient movement of sand grains on the sea floor in the new
Supplemental Materials section can be deleted. There is no need to suspend or move
sediment for anchor ice to entrain it.

**Line 177:** Zircon density is immaterial for ice-rafting as heavy grains are just as easily
entrained by anchor ice as light grains.

**Reply:** We appreciate the reviewer’s comment and understand the concern. We would
like to clarify, however, that the discussion of incipient motion in the Supplementary
Materials is not intended to assess how anchor ice entrains sediment, but rather to
evaluate whether coarse particles (e.g., >250 μm sand or >150 μm zircons) could have
been transported to the continental shelf surface **by rivers or currents** prior to any
potential ice interaction. Since sea ice typically forms and incorporates sediments from
shelf areas, the availability of coarse grains on the shelf surface is a prerequisite for any
entrainment by anchor ice. If such coarse particles cannot be delivered to the shelf under
prevailing or past hydrodynamic conditions, then their incorporation by anchor ice
becomes highly unlikely. Therefore, we think that the discussion on sediment mobility
remains relevant for evaluating the plausibility of sea ice as a transporter of coarse IRD.

To further clarify this point, we have revised the relevant paragraph in the main text as
follows:

“Moreover, the much coarser zircon grains found in the sand-rich IRD layers of cores
LV90-8-1 and LV90-9-1, compared to surface samples from the surrounding
continental shelves (Fig. 6), provide further evidence against sea ice as their transporter.
Zircon, being one of the densest minerals, is more difficult to transport **by rivers or**
**currents** than lighter minerals such as quartz or feldspar³⁹. Considering that even
lighter minerals >250 μm are rarely transported to the shelf (Fig. 5), theoretical
estimates suggest that the equivalent threshold for zircons is \sim 150 μm (Supplementary
Text S3), as shown by the absence of zircons >150 μm in modern surface sediments
(Fig. 6). **Therefore, sea ice, which incorporates seafloor particles through the**
**anchor ice mechanism, lacks zircons >150 μm .** In contrast, \sim 6% to \sim 18% of zircons
in the IRD layers exceed this size (Supplementary Fig. S8), implying that these coarse
and dense grains likely originated from more energetic transport processes—most
plausibly iceberg rafting.”

**Comment 5:** It is interesting that only total organic carbon is used for radiometric dates
in the two studied cores. TOC dates have been shown to have large offsets with
calcareous shell ^{14}C dates (Hansen et al., 2022, Quat. Geochronology). Radiocarbon is
good only back to about MIS 4 anyway, so not much help with MIS 6. Mn stratigraphy
is fine but it has its issues with accuracy because minor excursions to cold conditions
can deposit a grey layer mistaken for a full-blown glacial. However, despite these issues
the stratigraphy of the two cores is about as good as it will ever get given the data at
hand.

**Reply:** We agree with the reviewer that establishing accurate chronologies for Arctic
sediments remains a major challenge. The use of total organic carbon (TOC) for
radiocarbon dating in our study is indeed a compromise, as calcareous shells or other
biogenic materials are extremely scarce in deep Arctic marine sediments. As suggested
by Reviewer 1, we have now compiled all available chronological information into Fig.
3, including stratigraphic correlations, radiocarbon ages, OSL dates, and ^{230}Th data.
This figure provides a comprehensive overview of the age model for our cores and
highlights the associated uncertainties.

**Specific lines that could improve the paper:**

**Line 46:**icebergs calved from ice sheets as well as sediment entrained by anchor
ice across the Arctic (8 + new reference)

**Line 48:** ...by ice sheets or sea ice.

**Reply:** We appreciate the reviewer's suggestion for a more rigorous expression.
However, since our study focuses specifically on the reconstruction of the East Siberian
Ice Sheet, introducing sea ice in the introduction may be distracting to the reader. After
balancing precision and contextual clarity, we have revised the sentence to:

"Ice sheets across the Arctic produce large amounts of IRD that are subsequently
transported by icebergs⁸. As icebergs melt, they release their IRD load, depositing it on

the ocean floor⁹. Consequently, the **iceberg-carried IRD** potentially contains unique
provenance signals that reflect erosion and sediment production by ice sheets on the
Arctic continents and continental shelves.” in *Page 3, Line 46-50*.

A clearer distinction between iceberg and sea-ice transport could be better addressed in
the Discussion section, where more detailed context has been provided.

**Line 49-51:** ...except for Fe grain fingerprinting...

**Line 62:** Fe grain fingerprinting is also very effective in distinguishing E Siberian
sediment from other circum-Arctic sources.

**Reply:** According to your suggestion, we have added a discussion of Fe grain
fingerprinting in the revised manuscript and acknowledge that it is indeed effective in
distinguishing East Siberian sources. However, as discussed in our first-round revision
and in Supplementary Text 1, studies applying this provenance tool in both the central
Arctic and Fram Strait have shown minimal sediment contributions from East Siberia
(Darby and Zimmerman, 2008; Darby et al., 2017). Therefore, in the revised manuscript,
we shifted the emphasis from the “lack of suitable provenance indicators for East
Siberia” to the “lack of East Siberian signals in sediment records,” which we think more
accurately reflects the current state of the evidence. we have revised the relevant
paragraph in the main text as follows in *Page 3-4, Line 51-71*:

“The Arctic Ocean has received substantial IRD influxes during past glacial-interglacial
cycles⁸. However, identifying the precise sources of these IRD remains challenging due
to the difficulty of distinguishing debris provenance across the circum-Arctic
continental shelves (Fig. 1). Provenance proxies, such as mineral assemblages and
isotope geochemistry, have been effective in differentiating sediment sources between
the continental shelves of North America and the entire Eurasia (Fig. 1 and
Supplementary Fig. S1)^{1, 10, 11}; however, these methods struggle to distinguish detritus

from the continental shelves of eastern Eurasia (i.e. East Siberia) versus western-central
Eurasia (Supplementary Figs S1, S2 and S3; see details in Supplementary Text S1).
Although Fe oxide indicators have proven capable of identifying East Siberian
sources¹², their application has not revealed a significant East Siberian signature in
Arctic Ocean sediments¹³.

This absence of a clear East Siberian signal in the sedimentary record has limited our
understanding of Northern Hemisphere glaciation dynamics during past glacial-
interglacial cycles. Previous provenance studies of IRD have identified iceberg surges
originating from ice sheets on Eurasian or North American continental shelves in the
central and eastern Arctic Ocean (including the Lomonosov Ridge, Makarov Basin,
Amundsen Basin and Nansen Basin; Fig. 1) during the Quaternary glacial periods^{2, 4, 5,}
546 ^{7, 14}. In the western Arctic Ocean (including the Canada Basin and the Chukchi
Borderland; Fig. 1), sediments have received substantial quantities of IRD from the
North American Ice Sheet via the Beaufort Gyre^{1, 6, 15, 16}. However, the lack of evidence
for an East Siberian contribution to IRD continues to hinder the reconstruction of glacial
ice extent in this region. Whether and when an East Siberian ice sheet existed has
remained a topic of debate for over two decades (Fig. 1)¹⁷⁻²⁰

*Darby, D. A., and P. Zimmerman (2008), Ice-rafted detritus events in the Arctic during*
*the last glacial interval, and the timing of the Innuitian and Laurentide ice sheet*
*calving events, Polar Research, 27(2), 114-127.*

*Darby, D. A., J. T. Andrews, S. T. Belt, A. E. Jennings, and P. Cabedo-Sanz (2017),*
*Holocene Cyclic Records of Ice-Rafted Debris and Sea Ice Variations on the East*
*Greenland and Northwest Iceland Margins, Arctic, Antarctic, and Alpine Research,*
*49(4), 649-672.*

**Line 67:** Zircon ages should be touted as an additional tool not an end-all method for

provenance.

**Reply:** Thank you for this important comment. We agree that detrital zircon U-Pb
dating should be presented as a complementary provenance tool rather than a definitive
method. In response, we have revised the sentence on *Page 4, Line 81-82* to adopt a
more balanced tone. Specifically, we changed:

“More importantly, the age distribution of detrital zircons in sediments has the potential
to accurately distinguish between source regions with relatively small age differences
(e.g., a few tens of millions of years), offering an advantage over traditional provenance
tools, such as mineral assemblages and geochemical isotopes (e.g., Sr-Nd isotopes),
which often produce mixed provenance signals^{10,11}.”

to

“Compared to mineral assemblages and geochemical isotopes (e.g., Sr-Nd isotopes),
the age distribution of detrital zircons in sediments has the potential to accurately
distinguish source regions with relatively small age differences (e.g., a few tens of
millions of years)^{10,11}.”

**Line 92:** The discussion of the Pechora River, Kara Sea etc zircon sources belongs in
the Supplemental Materials. This entire paragraph could be eliminated and make the
paper more readable and shorter. The following paragraph states that the zircon ages
for the E Siberia area are unique and this suffices along with the existing Figure
showing this. Besides it is already stated that the zircon sources for these other areas is
discussed in Supplementary Materials.

**Reply:** In response your suggestion, we have streamlined this section by removing the
first paragraph and relocating the detailed discussion of zircon sources from the Pechora
River, Kara Sea, and other regions to the Supplemental Materials. The revised
paragraph now reads as follows:

“The zircon age distributions in these surface samples show regional distinctions (Fig.
2). Notably, a distinctive zircon age peak at ~90-110 Ma clearly fingerprints the East
Siberian provenance (Figs. 2M-U) (zircon age peaks in other continental shelves are
discussed in Supplementary Text S2). This age peak is found in sediments from the East
Siberian continental shelf (including the East Siberian Sea and Chukchi Sea) but absent
in sediments from the western-central Eurasian (including the Barents Sea, Kara Sea
and Laptev Sea) and North American continental shelves, highlighting the difference
between these areas (Fig. 2). The ~90-110 Ma zircons are likely derived from the
Okhotsk-Chukotka Volcanic Belt (~106–76 Ma) that is exposed in East Siberia (Fig.
1)²⁷. Although sediments from the Bering Sea also contain a 90–110 Ma peak (Fig. 2V),
their unique ~60 Ma age peak—likely associated with magmatic belts in the Yukon
River basin (Fig. 1)—distinguishes them from those of the East Siberian continental
shelf (Figs. 2M-V). Therefore, an age distribution characterized by a peak at ~90–110
600 Ma, with the absence of a ~60 Ma peak, serves as an exclusive provenance indicator
for East Siberia, distinguishing it from western-central Eurasia, North America, and the
Bering Sea.”

**Line 212:** Iceberg transport is not the only plausible explanation but one of the most
likely.

**Reply:** Thank you. We have changed this sentence to “Therefore, we propose that
iceberg was the most likely transport mechanism for the sand-rich IRD of LV90-8-1
and LV-9-1, particularly for coarse sands >250 μm and zircon grains >150 μm .” On
*Page 9, Line 204-205* according to your suggestion.

**Line 226:** Hopefully no one still proposes that a 1000m thick ice sheet covered the
entire Arctic Ocean. But a third possibility instead of this absurd kilometer thick ice
sheet would be that anchor ice entrained coarse sediment from the low-stand sea level

of glacial conditions. It is very feasible that during dropping sea level river gradients
increased and transported sand to the shoreline even to the shelf margin.

**Reply:** We appreciate the reviewer’s comment. This paragraph is primarily intended to
reflect the controversial history of the East Siberian Ice Sheet. Including the hypothesis
of a “~1000 m thick ice shelf” is to illustrate the range of scientific views that have
contributed to the ongoing debate. As for the alternative explanation involving anchor
ice, we respectfully suggest that it may be less relevant here, as the focus is on ice sheet
extent rather than sediment transport mechanisms. We have instead included the
discussion of anchor ice and sediment entrainment in the first paragraph of the
Discussion section, where it is more contextually appropriate.